# DAGER: Exact Gradient Inversion for Large Language Models

**Ivo Petrov**[*1], **Dimitar I. Dimitrov**[*1,2], **Maximilian Baader**[2],
**Mark Niklas Müller**[2,3], **Martin Vechev**[1,2]

[1] INSAIT, Sofia University "St. Kliment Ohridski"    [2] ETH Zurich    [3] LogicStar.ai

{ivo.petrov, dimitar.iliev.dimitrov}@insait.ai [1]
{mbaader, mark.mueller, martin.vechev}@inf.ethz.ch [2]

## Abstract

Federated learning works by aggregating locally computed gradients from multiple clients, thus enabling collaborative training without sharing private client data. However, prior work has shown that the data can actually be recovered by the server using so-called gradient inversion attacks. While these attacks perform well when applied on images, they are limited in the text domain and only permit approximate reconstruction of small batches and short input sequences. In this work, we propose DAGER, *the first algorithm to recover whole batches of input text exactly*. DAGER leverages the low-rank structure of self-attention layer gradients and the discrete nature of token embeddings to efficiently check if a given token sequence is part of the client data. We use this check to exactly recover full batches in the honest-but-curious setting without any prior on the data for both encoder- and decoder-based architectures using exhaustive heuristic search and a greedy approach, respectively. We provide an efficient GPU implementation of DAGER and show experimentally that it recovers full batches of size up to 128 on large language models (LLMs), beating prior attacks in speed (20x at same batch size), scalability (10x larger batches), and reconstruction quality (ROUGE-1/2 > 0.99).

## 1 Introduction

While large language models (LLMs) have demonstrated exceptional potential across a wide range of tasks, training them requires large amounts of data. However, this data is sensitive in many cases, leading to privacy concerns when sharing it with third parties for model training. Federated learning (FL) has emerged as a promising solution to addressing this issue by allowing multiple parties to collaboratively train a model by sharing only gradients computed on their private data with the server instead of the data itself. In particular, FL has been used to finetune LLMs while protecting private data [1, 2, 3] in privacy-critical domains, such as law [4] and medicine [5].

**Gradient Inversion Attacks** Unfortunately, recent work has shown that this private data can be recovered from the shared gradients using so-called gradient inversion attacks, raising concerns about the privacy guarantees of federated learning [6]. While most prior work on gradient inversion attacks has focused on image data [7, 8, 9], first works have demonstrated that text can also be recovered [6, 10, 11]. However, as these approaches are optimization-based, the discrete nature of text data poses a major challenge by inducing much harder optimization problems and limiting them to approximate recovery of small batch sizes and short sequences. Therefore, applying existing attacks methods on modern LLMs would be computationally infeasible or yield subpar reconstructions.

---

[*]Equal contribution.

38th Conference on Neural Information Processing Systems (NeurIPS 2024).

**This Work: Exact Recovery of Large Batches and Long Sequences** To overcome these limitations, we propose DAGER (**D**iscreteness-Based **A**ttack on **G**radients for **E**xact **R**ecovery), the first exact gradient inversion attack for (transformer-based) LLMs in the challenging honest-but-curious setting. Our key insight is that while discrete inputs pose a challenge for optimization-based attacks, they can be leveraged in combination with the low-rank structure of gradients to enable exact recovery via search-based attacks. Crucially, we show that the gradients of self-attention projection matrices in transformers are i) typically low-rank and ii) linear combinations of input embeddings. This allows us to check whether a given input embedding lies within the span of the gradient and was thus part of the input sequence. We use this to first recover the set of input tokens and then reconstruct the full sequences. For decoder architectures DAGER leverages their causal attention masks for to derive an efficient greedy recovery, while for encoder architectures, DAGER uses several heuristics to make exhaustive search tractable. As DAGER only requires propagating inputs through the first transformer block instead of full gradient computations, it scales to very large models. In fact, the higher internal dimension of these models even allows DAGER to recover more information as our low-rankness assumptions hold for larger batch sizes and longer sequences. We note that this approach is applicable both to the easier next-token prediction and to the harder classification setting.

**Evaluation** We demonstrate in an extensive evaluation that DAGER enables the exact recovery of long sequences and large batch sizes for both encoder- and decoder-based architectures, beating prior attacks in terms of speed (20x at same batch sizes), scalability (10x larger batches), and reconstruction quality (ROUGE-1/2 > 0.99). In particular, we show this for GPT-2 [12], LLaMa-2 [13], and BERT [14] across CoLA[15], SST-2 [16], Rotten Tomatoes [17] and ECHR [18], for batch sizes up to 128. Additionally, we demonstrate that DAGER is versatile and can be applied to a wide range of settings, including FedAvg [19], LoRA [20] finetuning and model quantization [21].

**Key Contributions** Our main contributions are:

- We show how the low-rankness of self-attention layer gradients can be leveraged to check whether specific discrete inputs were present in the input (Sec. 4).
- We leverage this key insight to propose DAGER, the first exact gradient inversion attack for transformers (Sec. 5).
- We conduct an extensive empirical evaluation demonstrating that DAGER is not only able to reconstruct inputs exactly but also scales to much larger batch sizes, longer input sequences, and larger models than prior attacks, while also being significantly faster to mount (Sec. 6).
- We provide an efficient GPU implementation of DAGER, that can be publicly accessed at https://github.com/insait-institute/dager-gradient-inversion.

## 2 Related Work

Gradient leakage attacks, first introduced by Zhu et al. [6], generally fall into two categories — honest-but-curious attacks [6, 22, 23, 7, 8, 24, 9, 10, 11, 25, 26, 27, 28], where the attacker passively observes the client's federated learning updates and tries to recover the data solely based on them, and malicious server attacks [29, 30, 31, 32, 33] where the attacker is further allowed to modify the federated learning model shared with the client. In this work, we focus on the harder to attack and more realistic honest-but-curious setting. A large body of the gradient leakage literature in this setting focuses on image data [7, 8, 24, 9, 27]. Differently, gradient leakage in the text domain remains successful only in the case of a malicious adversary [31, 32, 34]. In the honest-but-curious setting, the results either remain limited to short sequences and small batch sizes $B$ [6, 10, 11], require large number of gradient updates [25], or cannot recover the order of tokens in client sequences Xu et al. [35]. Further, state-of-the-art attacks require strong data priors [11, 25], and do not scale to realistic decoder-based LLMs. In contrast, DAGER, works on large batches and sequences for both encoder- and decoder-based transformers, including LLaMa-2 [13]. Additionally, unlike prior work, our attack works on both token prediction tasks and the harder setting of sentiment analysis [11] where label recovery methods, such as [35], are not applicable. Finally, DAGER has no requirements for the state of the model training. In instance, [25] exploits model memorization of the data, unlike DAGER, which can handle the more realistic setting of being applicable at any point in time. Further, in contrast to [26, 25, 31], we do not require the gradient of the embedding layer, making our setting significantly harder.

Table 1: Table of notations used in the technical description of DAGER.

| Symbol | Definition | Symbol | Definition |
|---|---|---|---|
| $B$ | Batch size | $\mathcal{L}$ | Loss function used for training |
| $P$ | Transformer context length | $d$ | Hidden(embedding) dimension |
| $L$ | Number of transformer blocks | $\mathcal{V}$ | Vocabulary set |
| $V$ | Vocabulary size $|\mathcal{V}|$ | $n_j$ | Token length for the $j$-th sequence |
| $n$ | $\max_j b_j$ - the length of the longest sequence | $b$ | $\sum_{j=1}^{B} n_j$ - the total number of non-padding tokens |
| $f^0$ | Embedding function (maps tokens to embeddings) | $z^{ij}$ | The $j$-th entry of the $i$-th position token's embedding. |
| $\boldsymbol{Z}^l$ | Input to the $l$-th attention layer | $\boldsymbol{M}$ | The attention mask |
| $\boldsymbol{W}_l^{\{Q,K,V\}}$ | Query/key/value projection weights for the $l$-th attention layer | $\{Q,K,V\}_l$ | The query/key/value embeddings in the $l$-th attention layer |
| $f_i^l$ | The $i$-th token embedding after the $l$-th transformer block | $\mathcal{T}_i^*$ | The set of client tokens at position $i$ |
| $\mathcal{S}_i^*$ | The set of batch sequences up to position $i$ | $s_1, s_2, ..., s_P$ | A sample sequence of $P$ tokens |
| $S_{best}^*$ | The set of the best reconstructed sequences | $\mathcal{D}^*$ | The set of distances to the span for each token/sequence |
| $\tau_l^{rank}$ | The singular value threshold for determining the rank of the l-th layer | $\tau_l$ | The distance threshold for filtering token candidates on the l-th layer |

While most prior honest-but-curious attacks leverage optimization methods to approximately recover the client inputs [6, 23, 7, 8, 24, 9, 10, 11], several works have shown that exact reconstruction is possible for batch size $B = 1$ under various conditions for different architectures [22, 26, 27]. Crucially, Dimitrov et al. [28] recently showed that $B > 1$ exact reconstruction from gradients of fully-connected layers is also possible. Our work, builds upon this result to show that exact gradient leakage is also possible for transformer-based LLMs.

## 3 Background and Notation

In this section, we introduce the background and notation required to understand our work. To this end, we first recall the basic operation of the transformer architecture in the context of LLMs, and then describe the result, first introduced in Dimitrov et al. [28] for linear layers, in the context of a self-attention layer showing that the gradients of its linear transformations have a low-rank structure. The notations used throughout this paper are summarized in Table 1 for clarity and ease of reference.

### 3.1 Transformers

In this paper, we consider LLMs based on both encoder and decoder transformer architectures trained using the FedSGD [36] protocol and a loss function $\mathcal{L}$. While we mainly focus on the harder-to-attack binary-classification loss typically used for sentiment analysis, we demonstrate that DAGER is equally applicable to the next-token prediction loss in Sec. 6, which contains more gradient information, as suggested by prior work Zhu et al. [6]. We denote the transformer's context length with $P$, its hidden dimension with $d$, the number of transformer blocks with $L$, the vocabulary of the tokenizer with $\mathcal{V}$, and its size with $V$. We present our approach for single-headed self-attention but it can be directly extended to multi-head self-attention, and we experimentally apply DAGER in this context.

**Transformer Inputs** We consider inputs batches consisting of $B$ sequences of tokens, where $n_j$ is the length of the $j^{\text{th}}$ batch element. Sequences with length $< n = \max_j(n_j)$ are padded. We denote the total number of non-padding tokens in a batch with $b = \sum_{j=1}^{B} n_j$.

**Token Embeddings** The discrete input tokens are usually embedded via a function $f^0 : [V] \times [P] \rightarrow \mathbb{R}^d$ mapping a token's vocabulary index $v$ and its position $i$ in the sequence to an embedding vector $\boldsymbol{z}^{ij} = f^0(v, i)$. These embeddings $\boldsymbol{z}^{ij}$ are then stacked row-wise to form the input $\boldsymbol{Z}_1 \in \mathbb{R}^{b \times d}$ to the first self-attention layer. Note that $f^0$ is known to the server, as it is part of the model. Further, while $f^0$ differs between models, typically it maps token indices to embedding vectors before optionally adding a positional encoding and applying a LayerNorm. Crucially, $f^0$ is applied per-token.

**Self-Attention** The stacked embeddings $\boldsymbol{Z}_1$ are then passed through a series of self-attention layers. We denote the input to the $l^{\text{th}}$ self-attention layer as $\boldsymbol{Z}_l \in \mathbb{R}^{b \times d}$, for $1 \le l \le L$. A self-attention layer is a combination of three linear layers: The query $\boldsymbol{Q}_l = \boldsymbol{Z}_l \boldsymbol{W}_l^Q$, key $\boldsymbol{K}_l = \boldsymbol{Z}_l \boldsymbol{W}_l^K$, and value $\boldsymbol{V}_l = \boldsymbol{Z}_l \boldsymbol{W}_l^V$ layer, which are then combined to compute the self-attention output:

$$\text{attention}(\boldsymbol{Q}_l, \boldsymbol{K}_l, \boldsymbol{V}_l) = \text{softmax}\left(\boldsymbol{M} \odot \frac{\boldsymbol{Q}_l \boldsymbol{K}_l^T}{\sqrt{d}}\right) \boldsymbol{V}_l,$$

where $\boldsymbol{M}$ is the binary self-attention mask, $\odot$ is the element-wise product, and the softmax is applied row-wise. $\boldsymbol{M}$ is chosen to ensure that padding tokens do not affect the layer's output. Further, for decoders, $\boldsymbol{M}$ ensures that only preceeding tokens are attended. For notational convenience, we denote as $f_i^l \colon \mathcal{V}^P \to \mathbb{R}^d$ the function that maps any sequence of input tokens to the $i^{\text{th}}$ input embedding at the $1 \le l \le L$ self-attention layer. Note that $f_i^l$ is part of the model and, thus, known to the attacker.

### 3.2 Low-Rank Decomposition of Self-Attention Gradients

For a linear layer $\boldsymbol{Y} = \boldsymbol{X}\boldsymbol{W} + (\boldsymbol{b}|\ldots|\boldsymbol{b})^T$ with a weight matrix $\boldsymbol{W} \in \mathbb{R}^{n \times m}$, a bias $\boldsymbol{b} \in \mathbb{R}^m$, and batched inputs $\boldsymbol{X} \in \mathbb{R}^{b \times n}$ and outputs $\boldsymbol{Y} \in \mathbb{R}^{b \times m}$, Dimitrov et al. [28] show that:

**Theorem 3.1** (Adapted from Dimitrov et al. [28]). *The network's gradient w.r.t. the weights $\boldsymbol{W}$ can be represented as the matrix product:*

$$\frac{\partial \mathcal{L}}{\partial \boldsymbol{W}} = \boldsymbol{X}^T \frac{\partial \mathcal{L}}{\partial \boldsymbol{Y}}. \tag{1}$$

*Further, when the batch size $b \le n, m$, the rank of $\frac{\partial \mathcal{L}}{\partial \boldsymbol{W}}$ is at most $b$.*

The rank limit follows directly from the dimensionalities of $\frac{\partial \mathcal{L}}{\partial \boldsymbol{Y}} \in \mathbb{R}^{b \times m}$ and $\boldsymbol{X} \in \mathbb{R}^{b \times n}$ in Eq. 1.

In this work, we apply Theorem 3.1 to the linear projection matrices $\boldsymbol{W}_l^{\{Q,K,V\}} \in \mathbb{R}^{d \times d}$. As long as the total number of tokens $b < d$, it states that the gradients $\frac{\partial \mathcal{L}}{\partial \boldsymbol{W}_l^Q}, \frac{\partial \mathcal{L}}{\partial \boldsymbol{W}_l^K}$, and $\frac{\partial \mathcal{L}}{\partial \boldsymbol{W}_l^V}$ are rank-deficient. Without loss of generality, for the rest of the paper we use $\frac{\partial \mathcal{L}}{\partial \boldsymbol{W}_l^Q} = \boldsymbol{Z}_l^T \frac{\partial \mathcal{L}}{\partial \boldsymbol{Q}_l}$ to explain our method.

## 4 Overview of DAGER

In this section, we provide a high-level overview of our method DAGER, illustrated in Fig. 1. DAGER is an attack that recovers the client input sequences from the shared gradients of a transformer-based LLM. DAGER works for both encoder and decoder-based LLMs, however, for simplicity here we focus on decoder-only LLMs. While, in theory, one could enumerate all possible batches of input sequences, and check whether they produce the desired gradients, this is infeasible in practice as it requires computing $V^{P \times B}$ dif-

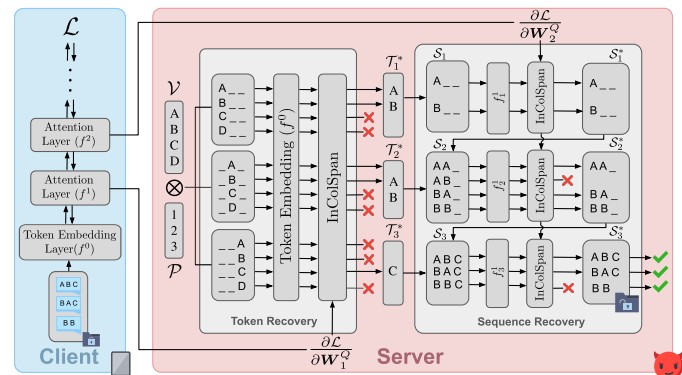

Figure 1: Overview of DAGER. DAGER first recovers the sets of client tokens $\mathcal{T}_i^*$ at each position $i \in \mathcal{P}$ by testing each token in the vocabulary $\mathcal{V}$ via a span check based on the client gradients of the first self-attention. Then it recursively combines them into partial client sequences $\mathcal{S}_i$ with length up to $i$, filtered to obtain the correct sequences $\mathcal{S}_i^*$ via the gradients of the second self-attention.

ferent gradients. We reduce the search space by leveraging the rank-deficiency of $\frac{\partial \mathcal{L}}{\partial \boldsymbol{W}_l^Q}$, discussed in Sec. 3.2, combined with the finite number of possible inputs to each self-attention corresponding to one of the $V^{P \times B}$ gradients above. For the rest of the section, we assume rank-deficiency of $\frac{\partial \mathcal{L}}{\partial \boldsymbol{W}_l^Q}$, that is $b < d$. This assumption is in practice satisfied for reasonable input lengths and batch sizes.

**Leveraging the Rank Deficiency** As the gradient matrix $\frac{\partial \mathcal{L}}{\partial \boldsymbol{W}_l^Q}$ is rank-deficient, i.e. $b < d$, the columns of $\frac{\partial \mathcal{L}}{\partial \boldsymbol{W}_l^Q}$ form a subspace of $\mathbb{R}^d$ of dimension $b$. Further, under mild assumptions

(see Theorem 5.1), the embedding vectors forming $\boldsymbol{Z}_l$ are linear combinations of the columns of $\frac{\partial \mathcal{L}}{\partial \boldsymbol{W}_l^Q} = \boldsymbol{Z}_l^T \frac{\partial \mathcal{L}}{\partial \boldsymbol{Q}_l}$. It is unlikely that any incorrect embedding vector, part of one of the $V^{P \times B}$ incorrect inputs $\boldsymbol{Z}_l$, is part of $\text{colspan}(\frac{\partial \mathcal{L}}{\partial \boldsymbol{W}_l^Q}) \subset \mathbb{R}^d$, as the hypervolume of this subspace is 0.

**Filtering Incorrect Embeddings**   We can efficiently filter out all incorrect client embeddings at any layer $l$ without computing their gradient, simply by checking if they are in $\text{colspan}(\frac{\partial \mathcal{L}}{\partial \boldsymbol{W}_l^Q})$. However, applying this procedure naively still requires us to check all $V^{P \times B}$ different client batches. Instead, we leverage this filtering in a two-stage recovery algorithm that first recovers the client tokens $\mathcal{T}_i^*$ at position $i$ using the rank deficiency of $\frac{\partial \mathcal{L}}{\partial \boldsymbol{W}_1^Q}$ (Token Recovery in Fig. 1), and then recovers the client batch of sequences $\mathcal{S}^*$ based on $\mathcal{T}_i^*$ and the rank deficiency of $\frac{\partial \mathcal{L}}{\partial \boldsymbol{W}_2^Q}$ (Sequence Recovery in Fig. 1).

**Token Recovery**   Our token recovery method relies on the observation that $f^0$ is computed per-token. Therefore, the input embeddings in $\boldsymbol{Z}_1$ are always part of the set $\{f^0(v, i) | v \in [V], i \in [P]\}$. We apply our span check above to this set for the first layer gradients $\frac{\partial \mathcal{L}}{\partial \boldsymbol{W}_1^Q}$ to filter the incorrect embeddings and their corresponding client tokens $v$ at position $i$, thus, constructing the set of correct client tokens $\mathcal{T}_i^*$ at position $i$.

**Sequence Recovery**   In our sequence recovery, we leverage the fact that $f_i^1$ is computed per-sequence and that the decoder mask $\boldsymbol{M}$ ensures that the second layer input embeddings at position $i$ do not depend on tokens with position $> i$, i.e., $f_i^1(s_1, \ldots, s_P) = f_i^1(s_1, \ldots, s_i)$, for any sequence of tokens $s_1, \ldots, s_P$. Crucially, for a correct client partial sequence $s_1, \ldots, s_{i-1}$ of length $i-1$ this allows us to find the correct next token in $\mathcal{T}_i^*$ by simply extending it with all possible token $\bar{s}_i \in \mathcal{T}_i^*$ and then checking which of the resulting embedding vectors $f_i^1(s_1, \ldots, s_{i-1}, \bar{s}_i)$ is correct, i.e, is in $\text{colspan}(\frac{\partial \mathcal{L}}{\partial \boldsymbol{W}_2^Q})$. We apply this procedure iteratively starting with the single token sequences $\mathcal{S}_1^* = \mathcal{T}_1^*$, extending them one token at a time to produce the partial sequence reconstructions $\mathcal{S}_i^*$, until the sequences cannot be extended anymore and return the result.

# 5   DAGER: Exact Sequence Recovery for Transformers

In this section, we present the technical details of DAGER. Specifically, we first theoretically derive of our filtering procedure based on the rank-deficiency of $\frac{\partial \mathcal{L}}{\partial \boldsymbol{W}_l^Q}$ in Sec. 5.1. We then describe how we apply it on the gradients of the first and second self-attention layers to respectively recover the client tokens (Sec. 5.2) and sequences (Sec. 5.3).

## 5.1   Efficient Embedding Filtering

Below, we discuss the technical details of our filtering procedure, outlined in Sec. 4, and prove its correctness. We first show that, under mild assumptions, the embedding vectors forming $\boldsymbol{Z}_l$ are linear combinations of the columns of $\frac{\partial \mathcal{L}}{\partial \boldsymbol{W}_l^Q}$, restating this in terms of $\text{rowspan}(\boldsymbol{Z}_l)$ and $\text{colspan}(\frac{\partial \mathcal{L}}{\partial \boldsymbol{W}_l^Q})$:

**Theorem 5.1.** *If $b < d$ and the matrix $\frac{\partial \mathcal{L}}{\partial \boldsymbol{Q}_l}$ is of full rank (rank $b$), then $\text{rowspan}(\boldsymbol{Z}_l) = \text{colspan}(\frac{\partial \mathcal{L}}{\partial \boldsymbol{W}_l^Q})$.*

Note that the assumption that $\frac{\partial \mathcal{L}}{\partial \boldsymbol{Q}_l}$ is full-rank holds in practice, as shown empirically in Dimitrov et al. [28], and that further $b < d$ is almost always satisfied, i.e., that the total number of tokens in the input is smaller than the internal dimensionality of the model, for practical LLMs. We discuss the assumptions in further detail in App. B.2. The latter then directly implies the rank-deficiency of $\frac{\partial \mathcal{L}}{\partial \boldsymbol{W}_l^Q}$, which we leverage to show:

**Theorem 5.2.** *When $b < d$, the probability of a random vector $\in \mathbb{R}^d$ to be part of $\text{colspan}(\frac{\partial \mathcal{L}}{\partial \boldsymbol{W}_l^Q})$ is almost surely $0$.*

Combining Theorems 5.1 and 5.2, we arrive at our main result stating that, if $b < d$, an embedding vector $z$ that is part of the client self-attention inputs $Z_l$ belongs to $\mathrm{colspan}(\frac{\partial \mathcal{L}}{\partial W_l^Q})$, while random embedding vectors that are not part of $Z_l$ almost surely do not.

**Span Check Implementation** While the above result holds under real, i.e., infinite precision, arithmetic, for our method to work in practice, we require an implementation that is both fast and robust to numerical errors caused by floating-point arithmetic. We, therefore, introduce the metric $d$, the distance between a candidate embedding vector $z$ and its projection on the $\mathrm{colspan}(\frac{\partial \mathcal{L}}{\partial W_l^Q})$:

$$d(z, l) = \| z - \mathrm{proj}(z, \mathrm{colspan}(\tfrac{\partial \mathcal{L}}{\partial W_l^Q})) \|_2. \qquad (2)$$

Intuitively, the closer $d(z, l)$ is to 0, the more likely $z$ is part of the span. To allow for efficient computation of the projection in Eq. 2, we first pre-compute an orthonormal basis for $\frac{\partial \mathcal{L}}{\partial W_l^Q}$ using an SVD and truncating the eigenvalues below a chosen threshold $\tau_l^{\mathrm{rank}}$. We can then trivially compute this projection, as the sum of projections onto each basis vector. Finally, we say that a vector $z$ is in $\mathrm{colspan}(\frac{\partial \mathcal{L}}{\partial W_l^Q})$, if the distance $d(z, l) < \tau_l$ is below a chosen per-layer threshold $\tau_l$.

## 5.2 Recovering Token Sets

We now describe how DAGER leverages the above filtering procedure to recover the input tokens exactly. To this end, we consider the set of all tokens in the model's vocabulary $v \in [V]$ at every possible position $i \in [P]$ and compute their input embeddings at the first layer via the per-token embedding function $f^0$. We then filter out token-position tuples $(v, i)$ whose embedding vectors $f^0(v, i)$ do not lie in $\mathrm{colspan}(\frac{\partial \mathcal{L}}{\partial W_1^Q})$ to obtain the set input tokens (across batch elements) at position $i$:

$$\mathcal{T}_i^* = \{ v \in [V] \mid d(f^0(v, i), 1) < \tau_1 \}. \qquad (3)$$

**Algorithm 1** Recovering Individual Tokens

1: **function** GETTOK($\frac{\partial \mathcal{L}}{\partial W_1^Q}, V, P, f^0, \tau_1$)
2:      $n \leftarrow 0$
3:      $\mathcal{T}_i^* \leftarrow \{\}, \mathcal{D}_i^* \leftarrow \{\}$
4:      **for** $v, i \leftarrow [V] \times [P]$ **do**
5:          $\bar{d} \leftarrow d(f^0(v, i), 1)$
6:          **if** $\bar{d} < \tau_1$ **then**
7:              $n \leftarrow \max(n, i + 1)$
8:              $\mathcal{T}_i^* \leftarrow \mathcal{T}_i^* + \{v\}$
9:              $\mathcal{D}_i^* \leftarrow \mathcal{D}_i^* + \{\bar{d}\}$
10:     **return** $n, \{\mathcal{T}_i^*\}_{i=0}^n, \{\mathcal{D}_i^*\}_{i=0}^n$

We formalize this process in Algorithm 1, where we simply enumerate all token position tuples $(v, i)$. Additionally, we compute the length of the longest input sentence $n$ as the largest position $i$ of any recovered tuple $(v, i)$ (Line 7). If $f^0$ is position-independent, e.g. when rotary instead of absolute positional embeddings are used, we recover the set of all input tokens $\mathcal{T}^* = \bigcup_i \mathcal{T}_i^*$ for every position. Our algorithm handles this at the sequence recovery stage at the price of slightly higher computational costs (see Theorem B.4).

While conceptionally simple, this approach is exceptionally effective and robust to the distance threshold $\tau_{1,2}$, as we demonstrate in Fig. 2 for the GPT-2 model [12] ($d = 768$) and a batch of $B = 32$ sequences consisting of $b = 391$ tokens. Despite the total number of tokens $b$ exceeding half the model dimensionality $d$, even our single layer (L1) filtering approach narrows the set of possible starting tokens, which are independent of the rest of the input, down to less than 300 from GPT-2's vocabulary of $V \approx 50K$ across a wide range of thresholds. Adding a second filtering stage using second-layer filtering, described next, allows DAGER to recover the exact 32 starting tokens. This is used to exactly narrow down the first token for each sequence, allowing us to inductively reconstruct the whole sentence for decoder-only models.

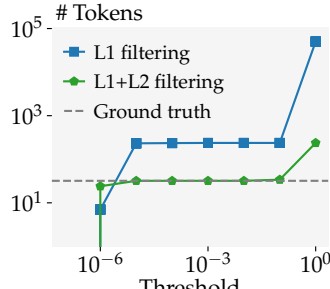

Figure 2: Effect of L1 and L2 Filtering

## 5.3 Recovering Sequences

Given the set of input tokens, recovered above, we now describe how to recover input sequences by applying our filtering procedure to the inputs of the second self-attention layer $Z_2$. We first define the set $\mathcal{S} = \mathcal{T}_1^* \times \cdots \times \mathcal{T}_P^*$ of all sequences formed using the recovered token sets $\mathcal{T}_i^*$. As the

second layer input embeddings $z_2 = f^1(s)$ are computed independently for each sequence $s$, one can naively enumerate all $s \in \mathcal{S}$, compute their second layer embedding vectors $f^1(s)$ and apply the span check for every position $i$ to obtain the true set of client sequences:

$$\mathcal{S}^* = \{s \in \mathcal{S} \mid d(f_i^1(s), 2) < \tau_2, \; \forall i \in [P]\}.$$

Unfortunately, this naive approach requires $\mathcal{O}(B^P)$ span checks. To alleviate this issue, we first show that the causal attention mask of decoder architectures allows us to greedily recover the exact sequences in polynomial time, before discussing heuristics that make an exhaustive search tractable for encoder-based architectures.

**Recovering Decoder Sequences**   Due to the causal attention mask $M$ in decoder architectures, the $i^{\text{th}}$ input of the second-self attention layer $f_i^1(s)$ depends only on the first $i$ tokens of the input sequence $s$. We can thus apply a span check on the results of $f_i^1$ to check arbitrary sequence prefixes of length $i$. We leverage this insight in Algorithm 2 to iteratively recover the sets $\mathcal{S}_i^*$ (Line 10) of input sequence prefixes

---

**Algorithm 2** DAGER for Decoders

1: **function** ATTDEC$(B, \frac{\partial \mathcal{L}}{\partial W_1^Q}, \frac{\partial \mathcal{L}}{\partial W_2^Q}, V, P, f^{0/1}, \tau_{1/2})$
2: $\quad n, \mathcal{T}^*, \mathcal{D}^* \leftarrow$ GETTOK$(\frac{\partial \mathcal{L}}{\partial W_1^Q}, V, P, f^0, \tau_1)$
3: $\quad \mathcal{S}_0^* \leftarrow \{\{\}\}_{j=1}^B$
4: $\quad$ **for** $i \leftarrow 1, \dots, n$ **do**
5: $\qquad$ TokenFound $\leftarrow$ False
6: $\qquad \mathcal{S}_i \leftarrow \mathcal{S}_{i-1}^* \times T_i^*$
7: $\qquad \mathcal{S}_i^* \leftarrow \{\}$
8: $\qquad$ **for** $s \in \mathcal{S}_i$ **do**
9: $\qquad\quad$ **if** $d(f_i^1(s), 2) < \tau_2$ **then**
10: $\qquad\qquad \mathcal{S}_i^* \leftarrow \mathcal{S}_i^* + \{s\}$
11: $\qquad\qquad$ TokenFound $\leftarrow$ True
12: $\qquad$ **if not** TokenFound **then**
13: $\qquad\qquad$ **break**
14: $\quad \mathcal{S}_{\text{best}}^* \leftarrow$ TOPUNIQUE$(\bigcup_{i=1}^l \mathcal{S}_i^*, \frac{\partial \mathcal{L}}{\partial W_2^Q}, B)$
15: $\quad$ **return** $\mathcal{S}_{\text{best}}^*$

---

$$\mathcal{S}_i^* = \{s \in \mathcal{S}_{i-1}^* \times \mathcal{T}_i^* \mid d(f_i^1(s), 2) < \tau_2\},$$

starting from the set of empty sequences $\mathcal{S}_0$ and extending them one token at a time (Line 6) until none of our sequences can be extended any further (Line 12).

For models with a small internal dimension $d$, or batches with a large number of total tokens $b$, the weight gradient $\frac{\partial \mathcal{L}}{\partial W^Q}$ might become full rank and all embeddings would pass our span check. To avoid this we set a maximum rank threshold, $\tilde{b} = \min(b, d - \Delta_b)$ for the orthonormal basis computed via SVD (See Sec. 5.1). We visualize the effect of different $\Delta_b$ on GPT-2, in Fig. 3, and observe that $\Delta_b = 20$ offers the best trade-off between stability and accuracy, yielding almost perfect reconstruction even for very large inputs with $b$ very close to $d$.

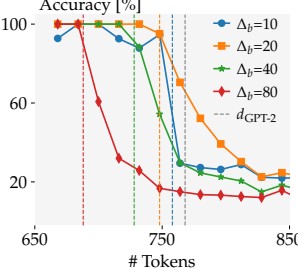

Figure 3: Encoder Ablation Study

**Recovering Encoder Sequences**   For encoders, all second-layer embeddings $f_i^1(s)$ depend on all input tokens. We thus cannot use the greedy reconstruction discussed above but have to enumerate all sequences in $\mathcal{S}$. To make this search tractable, we leverage the following heuristics. We can determine the positions $i$ of end-of-sequence (EOS) tokens in the input to determine the input sequence lengths $n_j$. This allows us to recover input sequences by increasing length and eliminate the tokens constituting the recovered sequences from the token sets $\mathcal{T}_i^*$. Additionally, we truncate the proposal token sets $\mathcal{T}_i^*$ to the batch size $B$ token closest to colspan$(\frac{\partial \mathcal{L}}{\partial W_1^Q})$. Finally, we

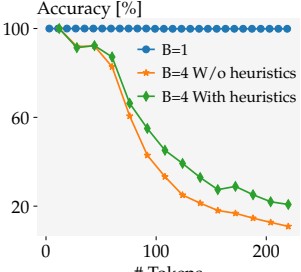

Figure 4: Encoder Ablation Study

always consider at most 10M sequences from $\mathcal{S}$ before returning the ones closest to colspan$(\frac{\partial \mathcal{L}}{\partial W_2^Q})$.

We demonstrate the effectiveness of our heuristics in Fig. 4 for the base BERT model and different batch sizes $B$ and note that for $B = 1$ we can still recover inputs perfectly, as $|\mathcal{T}_i^*| = 1$. We provide more details on DAGER for encoder-architectures in App. B.6.

# 6 Experimental Evaluation

We now describe our extensive experimental evaluation of DAGER. Our results demonstrate significant performance improvements compared to prior methods, on a variety of settings. We also present ablation studies for isolating the effects of each DAGER component.

## 6.1 Experimental Setup

We evaluate DAGER on both encoder- and decoder-based models including BERT [14], GPT-2 [12], and variations of LLaMa [13, 37]. We consider three sentiment analysis datasets – CoLA [15], SST-2 [16], and Rotten Tomatoes (RT) [17], featuring sequences of varying lengths between typically 4 and 27 words. Additionally, we consider the ECHR [18] dataset, which contains sentences exceeding 1000 words to demonstrate the scalability of our approach in sequence length. We provide a more detailed description of our architectures and datasets in App. C. Following previous work, we report the mean and error of the ROUGE-1/2 [38] scores, i.e., the overlap rate of unigrams and bigrams, respectively, over 100 batches, excluding padding tokens. We report the error as the 95% confidence interval given by twice the standard error. Wherever we cannot assume normality of the mean's distribution, we estimate the interval by generating $10\,000$ random samples by bootstrapping. Additional details regarding computational requirements and hyperparameters can be found in App. C.

**Comparison against baselines**    First, we compared our performance against the state-of-the-art algorithms TAG [10] and LAMP [11] with batch sizes ranging between $B = 1$ and $B = 8$. We run the two attacks for 2500 iterations per experiment, making use of the hyperparameters described in their respective studies. As Balunović et al. [11] provide two variations of the LAMP algorithm based on different objective functions — $\text{LAMP}_{L2 + L1}$ and $\text{LAMP}_{\text{Cos}}$, we only report results from the variation with the higher ROUGE-1 score. In Table 2, we show results on GPT-2$_{\text{BASE}}$ and BERT$_{\text{BASE}}$, assessing the performance on decoder-based and encoder-based models respectively.

The results indicate that for decoder-based models, such as GPT2, DAGER achieves near-perfect reconstructions across all datasets and batch sizes, significantly outperforming the baseline algorithms in every setting. Importantly, as further elaborated in App. C.1, DAGER achieves that while also being significantly more efficient — 100 batches of size 8 on RT took 3.5 hours vs TAG and LAMP which required $\approx 10$ and 50 hours, respectively. Additionally, we confirm the claims made by Balunović et al. [11] that LAMP outperforms TAG in the majority of settings. Examples of reconstructed sentences can be seen in Table 9 in App. C.3. We note that while we observe non-perfect ROUGE-2 scores on the SST-2 dataset, this is entirely due to an artifact of our metric library that assigns ROUGE-2 score of 0 to the SST-2's single-word sequences. We kept this behaviour to avoid having to rerun the baseline experiments, that also relied on this.

Further, Table 2 shows a significant improvement over prior work on encoder-based models like BERT, with near-perfect reconstruction for $B = 1, 2$, and an average of 43% more tokens recovered for larger batch sizes. A significant advantage of DAGER over the baselines is its ability to more accurately recover the sentence structure, as evidenced by the much higher ROUGE-2 scores.

**Main experiments**    While prior attacks' performances become very poor for batch sizes as little as 8, we now demonstrate that DAGER is only limited by the embedding dimension of the model. To this end, in Table 3 we compare two decoder-only models, GPT-2$_{\text{BASE}}$ with $d = 768$, and LLaMa-2 7B with $d = 4096$ on $B$ as large as 128.

The results are consistent with our claims that DAGER produces almost perfect reconstructions in all cases when the total number of client tokens is not extremely close to the embedding dimension $d$. Further, while on LLaMa-2 DAGER achieves near-perfect reconstructions even up to a batch size of 128, for GPT-2 DAGER shows partial or complete failure for $B = 64, 128$. This suggests that despite the significant computational costs of $> 2$ hours per batch for $B = 128$ on LLaMa-2, larger models have the potential to leak significantly more information. This is especially concerning given the current trend of ever-increasing model sizes. Finally, we observe the effect of attempting a best-effort reconstruction by establishing a rank threshold, as described in Sec. 5.3, when the gradients are of full rank. This allows DAGER to achieve a ROUGE-1 score of 30.3 (instead of 0) for GPT-2 on CoLA $B = 128$. A thorough ablation study on the advantage of this heuristic can be found in App. C.2.

Table 2: Comparison of sequence reconstruction from gradients between DAGER and the baseline algorithms TAG and LAMP on various batch sizes and datasets. R-1 and R-2 denote the ROUGE-1 and ROUGE-2 scores respectively.

| | | | $B=1$ | | $B=2$ | | $B=4$ | | $B=8$ | |
|---|---|---|---|---|---|---|---|---|---|---|
| | | | R-1 | R-2 | R-1 | R-2 | R-1 | R-2 | R-1 | R-2 |
| GPT-2 | CoLA | TAG | $7.0\pm2.5$ | $0.54\pm0.54$ | $8.0\pm2.0$ | $1.4\pm1.3$ | $7.8\pm1.2$ | $0.8\pm0.5$ | $5.3\pm0.7$ | $0.4\pm0.2$ |
| | | LAMP | $73.3\pm4.5$ | $43.3\pm7.0$ | $26.8\pm2.8$ | $11.0\pm3.0$ | $13.4\pm1.4$ | $3.9\pm1.2$ | $8.9\pm1.2$ | $1.9\pm0.6$ |
| | | DAGER | $\mathbf{100.0\pm0.0}$ | $\mathbf{100.0\pm0.0}$ | $\mathbf{100.0\pm0.0}$ | $\mathbf{100.0\pm0.0}$ | $\mathbf{100.0\pm0.0}$ | $\mathbf{100.0\pm0.0}$ | $\mathbf{100.0\pm0.0}$ | $\mathbf{100.0\pm0.0}$ |
| | SST-2 | TAG | $5.3\pm0.5$ | $0.0\pm0.0$ | $6.0\pm1.7$ | $0.5\pm0.4$ | $6.1\pm1.2$ | $0.6\pm0.6$ | $4.4\pm0.6$ | $0.2^{+0.6}_{-0.1}$ |
| | | LAMP | $62.2\pm6.9$ | $31.8\pm8.4$ | $21.4\pm3.1$ | $9.2\pm3.1$ | $9.8\pm2.0$ | $2.7\pm1.3$ | $8.1\pm1.1$ | $0.7\pm0.4$ |
| | | DAGER | $\mathbf{100.0\pm0.0}$ | $\mathbf{86.0\pm7.0}$ | $\mathbf{100.0\pm0.0}$ | $\mathbf{89.5\pm4.1}$ | $\mathbf{100.0\pm0.0}$ | $\mathbf{92.8\pm2.4}$ | $\mathbf{100.0\pm0.0}$ | $\mathbf{92.9\pm1.6}$ |
| | Rotten Tomatoes | TAG | $7.1\pm1.8$ | $0.1^{+0.4}_{-0.1}$ | $7.0\pm1.2$ | $0.1^{+0.2}_{-0.1}$ | $6.2\pm0.8$ | $0.1^{+0.2}_{-0.1}$ | $6.1\pm0.5$ | $0.1\pm0.1$ |
| | | LAMP | $31.4\pm4.4$ | $9.3\pm3.6$ | $11.2\pm1.2$ | $0.9\pm0.42$ | $6.3\pm1.1$ | $0.9\pm0.6$ | $6.8\pm0.7$ | $0.3^{+0.2}_{-0.1}$ |
| | | DAGER | $\mathbf{100.0\pm0.0}$ | $\mathbf{100.0\pm0.0}$ | $\mathbf{100.0\pm0.0}$ | $\mathbf{100.0\pm0.0}$ | $\mathbf{99.3^{+0.7}_{-1.7}}$ | $\mathbf{99.3^{+0.7}_{-1.8}}$ | $\mathbf{100.0^{+0.0}_{-0.1}}$ | $\mathbf{99.9^{+0.1}_{-0.6}}$ |
| BERT | CoLA | TAG | $78.9\pm4.4$ | $10.3\pm3.0$ | $68.9\pm4.2$ | $7.7\pm1.7$ | $56.3\pm3.4$ | $6.8\pm1.4$ | $45.9\pm1.9$ | $3.9\pm0.6$ |
| | | LAMP | $89.6\pm2.5$ | $51.9\pm6.7$ | $77.8\pm3.6$ | $31.5\pm4.6$ | $66.2\pm3.4$ | $21.8\pm1.7$ | $52.9\pm2.2$ | $13.1\pm1.9$ |
| | | DAGER | $\mathbf{100.0\pm0.0}$ | $\mathbf{100.0\pm0.0}$ | $\mathbf{100.0\pm0.0}$ | $\mathbf{100.0\pm0.0}$ | $\mathbf{94.0\pm2.0}$ | $\mathbf{89.9\pm3.1}$ | $\mathbf{67.8\pm2.3}$ | $\mathbf{48.8\pm4.5}$ |
| | SST-2 | TAG | $75.4\pm4.3$ | $19.0\pm6.9$ | $71.8\pm3.6$ | $16.0\pm3.9$ | $61.0\pm3.4$ | $12.3\pm2.8$ | $50.4\pm2.4$ | $9.2\pm1.6$ |
| | | LAMP | $88.8\pm3.0$ | $56.8\pm7.9$ | $82.4\pm3.6$ | $45.7\pm6.0$ | $69.5\pm3.6$ | $32.5\pm4.4$ | $56.9\pm2.6$ | $19.1\pm2.8$ |
| | | DAGER | $\mathbf{100.0\pm0.0}$ | $\mathbf{100.0\pm0.0}$ | $\mathbf{99.3^{+0.7}_{-2.0}}$ | $\mathbf{99.0^{+0.8}_{-2.1}}$ | $\mathbf{95.6\pm2.2}$ | $\mathbf{93.0\pm3.3}$ | $\mathbf{74.1\pm3.3}$ | $\mathbf{59.8\pm2.9}$ |
| | Rotten Tomatoes | TAG | $60.1\pm4.4$ | $3.3\pm1.2$ | $49.2\pm3.5$ | $3.0\pm0.9$ | $33.7\pm2.5$ | $1.6\pm0.7$ | $25.4\pm1.2$ | $0.9\pm0.4$ |
| | | LAMP | $64.7\pm4.4$ | $16.5\pm3.9$ | $46.4\pm3.7$ | $7.6\pm2.0$ | $35.1\pm2.7$ | $4.2\pm1.3$ | $27.3\pm1.4$ | $2.0\pm0.6$ |
| | | DAGER | $\mathbf{100.0\pm0.0}$ | $\mathbf{100.0\pm0.0}$ | $\mathbf{98.1\pm1.2}$ | $\mathbf{96.5\pm1.8}$ | $\mathbf{66.8\pm3.2}$ | $\mathbf{50.1\pm4.4}$ | $\mathbf{37.1\pm1.2}$ | $\mathbf{11.4\pm1.3}$ |

Table 3: Main experiments on the GPT-2$_{\text{BASE}}$ and LLaMa-2 (7B) models with higher batch sizes on various datasets. R-1 and R-2 denote the ROUGE-1 and ROUGE-2 scores respectively.

| | | $B=16$ | | $B=32$ | | $B=64$ | | $B=128$ | |
|---|---|---|---|---|---|---|---|---|---|
| | | R-1 | R-2 | R-1 | R-2 | R-1 | R-2 | R-1 | R-2 |
| CoLA | GPT-2 | $100.0\pm0.0$ | $100.0\pm0.0$ | $100.0\pm0.0$ | $100.0\pm0.0$ | $100.0\pm0.0$ | $100.0\pm0.0$ | $30.3\pm1.0$ | $14.6\pm0.9$ |
| | LLaMa-2 (7B) | $100.0\pm0.0$ | $100.0\pm0.0$ | $100.0\pm0.0$ | $100.0\pm0.0$ | $99.9^{+0.0}_{-0.1}$ | $99.9^{+0.0}_{-0.1}$ | $99.5\pm0.2$ | $99.3\pm0.3$ |
| SST-2 | GPT-2 | $100.0\pm0.0$ | $94.6\pm1.1$ | $100.0^{+0.0}_{-0.1}$ | $93.4\pm1.0$ | $92.9\pm3.0$ | $85.0\pm3.5$ | $13.7\pm1.4$ | $4.3\pm0.5$ |
| | LLaMa-2 (7B) | $100.0\pm0.0$ | $100.0\pm0.0$ | $99.9^{+0.0}_{-0.1}$ | $99.9\pm0.1$ | $99.9\pm0.1$ | $99.9\pm0.1$ | $98.2\pm0.4$ | $97.8\pm0.4$ |
| Rotten Tomatoes | GPT-2 | $100.0\pm0.0$ | $99.9^{+0.1}_{-0.3}$ | $98.0\pm1.7$ | $97.8\pm1.8$ | $2.8\pm1.1$ | $1.1\pm0.4$ | $0.0\pm0.0$ | $0.0\pm0.0$ |
| | LLaMa-2 (7B) | $100.0^{+0.0}_{-0.1}$ | $100.0^{+0.0}_{-0.1}$ | $100.0\pm0.0$ | $100.0\pm0.0$ | $97.9\pm0.5$ | $97.8\pm0.5$ | $99.7^{+0.1}_{-0.2}$ | $99.7^{+0.2}_{-0.3}$ |

**Reconstruction under FedAvg** The FedAvg algorithm [19] is among the most widely used protocols in federated learning. It features $E$ training epochs on minibatches of size $B_{mini} < B$ with a fixed learning rate $\eta$. Despite featuring multiple low-rank gradient updates, this setting it remains vulnerable to our attack, as we elaborate in App. B.3. We show in Table 4 that FedAvg is susceptible to gradient leakage under DAGER for a range of reasonable learning rates and number of epochs.

Table 4: Experiments on the FedAVG setting on the GPT-2 model with a batch size of 16 on the Rotten Tomatoes dataset. We use default set of hyperparameters of $E = 10$ epochs, learning rate $\eta = 10^{-4}$ and mini-batch size $B_{mini} = 4$. R-1 and R-2 denote ROUGE-1 and ROUGE-2 respectively.

| E | R-1 | R-2 | $\eta$ | R-1 | R-2 | $B_{mini}$ | R-1 | R-2 |
|---|---|---|---|---|---|---|---|---|
| 2 | $98.4\pm0.9$ | $98.0\pm1.0$ | $10^{-5}$ | $100.0^{+0.0}_{-0.2}$ | $99.8^{+0.2}_{-0.4}$ | 2 | $93.2\pm1.7$ | $92.3\pm1.9$ |
| 5 | $97.3\pm1.2$ | $96.8\pm1.3$ | $5\times10^{-5}$ | $99.8^{+0.2}_{-0.5}$ | $99.6^{+0.3}_{-0.7}$ | 4 | $95.4\pm1.6$ | $94.7\pm1.7$ |
| 10 | $95.4\pm1.6$ | $94.7\pm1.7$ | $10^{-4}$ | $95.4\pm1.6$ | $94.7\pm1.7$ | 8 | $98.6^{+0.5}_{-0.9}$ | $98.2^{+0.7}_{-1.0}$ |
| 20 | $96.0\pm1.4$ | $95.3\pm1.6$ | $5\times10^{-4}$ | $84.2\pm1.8$ | $82.2\pm1.9$ | 16 | $100.0\pm0.0$ | $99.8^{+0.2}_{-0.3}$ |

**Effect of Fine-tuning Methods** We further demonstrate DAGER's versatility across a range of pretraining paradigms, including quantized models and Low-Rank Adaptation (LoRA) [20] finetuning. For both LLaMa-3 70B with $B = 1$ and LLaMa-3.1 8B with $B = 32$ at 16-bit quantization, we observed excellent ROUGE-1 and ROUGE-2 scores (>99%) (see Table 11 in App. C.5). We also present near-exact reconstructions under LoRA training, as DAGER can be directly applied to the decomposed weight matrix, with further technical specifics detailed in App. B.4. With LoRA updates of rank $r = 256$, which is standard for the LLaMa-2 model as noted by Biderman et al. [39], we observe ROUGE-1 and ROUGE-2 scores in the region of $94 - 95\%$, given in Table 11. These results reaffirm that DAGER is applicable to common fine-tuning methods.

**Effect of Model Size and Training on Reconstruction** Prior work [11, 10] suggests that the size of a model, as well as, the degree of pre-training significantly affects the amount of leaked client information. To this end, in Table 5 we evaluate DAGER on the larger (GPT-2$_{\text{LARGE}}$) and

Table 5: Experiments on GPT-2 variations in different settings on the Rotten Tomatoes dataset. R-1 and R-2 denote the ROUGE-1 and ROUGE-2 scores respectively.

| | $B = 16$ | | $B = 32$ | | $B = 64$ | | $B = 128$ | |
|---|---|---|---|---|---|---|---|---|
| | R-1 | R-2 | R-1 | R-2 | R-1 | R-2 | R-1 | R-2 |
| GPT-2$_{\text{BASE}}$ | $\mathbf{100.0 \pm 0.0}$ | $\mathbf{99.9^{+0.1}_{-0.3}}$ | $98.0 \pm 1.7$ | $97.8 \pm 1.8$ | $2.8 \pm 1.1$ | $1.1 \pm 0.4$ | $0.0 \pm 0.0$ | $0.0 \pm 0.0$ |
| GPT-2$_{\text{FineTuned}}$ | $\mathbf{100.0 \pm 0.0}$ | $99.8^{+0.1}_{-0.3}$ | $96.4 \pm 2.3$ | $96.0 \pm 2.5$ | $0.84 \pm 0.6$ | $0.2^{+0.2}_{-0.1}$ | $0.0 \pm 0.0$ | $0.0 \pm 0.0$ |
| GPT-2$_{\text{NextToken}}$ | $99.9 \pm 0.0$ | $99.7^{+0.2}_{-0.3}$ | $99.6^{+0.3}_{-0.9}$ | $99.4^{+0.3}_{-0.9}$ | $2.3 \pm 0.8$ | $0.5^{+0.3}_{-0.2}$ | $0.0 \pm 0.0$ | $0.0 \pm 0.0$ |
| GPT-2$_{\text{LARGE}}$ | $\mathbf{100.0 \pm 0.0}$ | $99.8^{+0.1}_{-0.3}$ | $\mathbf{100.0 \pm 0.0}$ | $\mathbf{99.9^{+0.1}_{-0.2}}$ | $\mathbf{44.1 \pm 4.2}$ | $\mathbf{38.1 \pm 4.7}$ | $0.0 \pm 0.0$ | $0.0 \pm 0.0$ |

pre-trained for 2 epochs (GPT-2$_{\text{FineTuned}}$) variants of GPT-2 on the RT dataset for batch sizes up to 128. We observe very little difference in performance. In fact, the GPT-2$_{\text{LARGE}}$'s larger embedding dimension allows us to approximately reconstruct more tokens at larger batch sizes. We further note that a larger vocabulary does not negatively impact DAGER, as can be seen from the applications on LLaMa3.1-8B and LLaMa3.1-70B (see Table 11), which feature a vocabulary size of 128,256 tokens.

**Reconstruction under Next-Token Prediction** Additionally, we evaluate our model on the next-token prediction task to demonstrate DAGER's efficacy under different contexts. We again achieve near-perfect results with ROUGE-1/2 scores of $> 99$. DAGER does not reach perfect scores because the last token in each client sequence only acts as a target and it is, thus, masked out from the input.

**Reconstruction of Long Sequences** Finally, to demonstrate our robustness to long sequences, we conducted a single experiment with $B = 1$ on the ECHR dataset truncated to 512 tokens. We obtain a perfect score of $\mathbf{100.0 \pm 0.0}$ for ROUGE-1 and ROUGE-2 on GPT-2$_{\text{BASE}}$, emphasizing the general applicability of DAGER. In contrast, in the same setting LAMP achieves a ROUGE-1 of $10.1 \pm 2.3$.

# 7 Limitations

As discussed in Sec. 5 and demonstrated in Sec. 6, the performance of DAGER on decoder-based models is only constrained by the embedding dimension $d$. While an exact reconstruction for a number of tokens $b > d$ is unachievable, we showed that the attack's effectiveness decreases only gradually with $b$. Given our robust performance in an undefended setting, an interesting avenue for future work is to improve DAGER against different defense mechanisms, including but not limited to using the Differential Privacy SGD optimization process (DPSGD)[40].

On the other hand, applying DAGER on encoder-based architectures for larger batches ($B >> 8$) becomes challenging due to the high-order polynomial growth of the search space volume with respect to the batch size. These computational constraints make comprehensive exploration of the search space nearly impossible, thereby reducing the likelihood of achieving a feasible reconstruction. This issue extends to longer sequences, where the size of the search space expands exponentially with the maximum sequence length. To mitigate these effects, we propose that future research could focus on exploring further heuristics to efficiently reduce the search space.

# 8 Conclusion

We introduced DAGER, the first gradient inversion attack for transformers able to recover large batches of input text exactly. By exploiting the rank-deficiency of self-attention layer gradients and discreteness of the input space, we devised a greedy algorithm and a heuristic search approach for decoder-based and encoder-based architectures, respectively. Our results show that DAGER achieves exact reconstruction for batch sizes up to 128 and sequences up to 512 tokens. We further demonstrate DAGER's effectiveness across model sizes, architectures, degrees of pre-training, and federated learning algorithms, establishing the widespread applicability of our attack.

Our work demonstrates that recent decoder-based LLMs are particularly vulnerable to data leakage, allowing adversaries to recover very large batches and sequences in the absence of a robust defense mechanism. This underlying vulnerability highlights the need for increased awareness and development of effective countermeasures in privacy-critical applications. We hope this paper can facilitate further research into creating reliable frameworks for effective and private collaborative learning.

**Acknowledgments**

This research was partially funded by the Ministry of Education and Science of Bulgaria (support for INSAIT, part of the Bulgarian National Roadmap for Research Infrastructure).

This work has been done as part of the EU grant ELSA (European Lighthouse on Secure and Safe AI, grant agreement no. 101070617) . Views and opinions expressed are however those of the authors only and do not necessarily reflect those of the European Union or European Commission. Neither the European Union nor the European Commission can be held responsible for them.

The work has received funding from the Swiss State Secretariat for Education, Research and Innovation (SERI).

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

## A  Broader Impact

In this work, we demonstrate that it is possible to exactly reconstruct large batches of textual data from gradients in the honest-but-curious setting. Our findings are widely applicable across different transformer-based LLM architectures, showing that, in contrast to prior belief, language transformers are actually more susceptible to gradient leakage attacks than other architectures. While our work naturally substantially increases the privacy risks posed to federated learning clients training LLMs, we also believe that sharing our work is crucial for finding future solutions to the issues we uncover.

Importantly, we find that the recent decoder-based models are much more susceptible to gradient leakage attacks due to the causal nature of their self-attention masks. Our work implies that in the absence of proper defense mechanisms, receiving gradients from those models is essentially equivalent to receiving the client data directly. Further, we show that the attacker's ability to mount DAGER grows with the embedding size $d$, suggesting the privacy risks posed by DAGER in practical settings will only grow over time. With these considerations in mind, we emphasize the importance of providing privacy safeguards via secure aggregation, larger batch sizes, or gradient perturbations.

## B  Additional Technical Details of Our Method

### B.1  Deferred Proofs

**Theorem 5.1.** *If $b < d$ and the matrix $\frac{\partial \mathcal{L}}{\partial \boldsymbol{Q}_l}$ is of full rank (rank b), then* $\mathrm{rowspan}(\boldsymbol{Z}_l) = \mathrm{colspan}(\frac{\partial \mathcal{L}}{\partial \boldsymbol{W}_l^Q})$.

*Proof.* We split the proof into two parts. We first prove $\mathrm{colspan}(\frac{\partial \mathcal{L}}{\partial \boldsymbol{W}_l^Q}) \subseteq \mathrm{rowspan}(\boldsymbol{Z}_l)$ and then prove that $\mathrm{rank}(\frac{\partial \mathcal{L}}{\partial \boldsymbol{W}_l^Q}) = \mathrm{rank}(\boldsymbol{Z}_l)$, thus implying the two spaces are the same.

For the first part of the proof, we observe that due to the matrix multiplication in $\frac{\partial \mathcal{L}}{\partial \boldsymbol{W}_l^Q} = \boldsymbol{Z}_l^T \frac{\partial \mathcal{L}}{\partial \boldsymbol{Q}_l}$ all columns of $\frac{\partial \mathcal{L}}{\partial \boldsymbol{W}_l^Q}$ are linear combinations of the columns of $\boldsymbol{Z}_l^T$ with coefficients given by $\frac{\partial \mathcal{L}}{\partial \boldsymbol{Q}_l}$. Thus, $\mathrm{colspan}(\frac{\partial \mathcal{L}}{\partial \boldsymbol{W}_l^Q}) \subseteq \mathrm{colspan}(\boldsymbol{Z}_l^T) = \mathrm{rowspan}(\boldsymbol{Z}_l)$.

For the second part of the proof, we observe that since $\frac{\partial \mathcal{L}}{\partial \boldsymbol{Q}_l}$ is of rank $b$ and $\mathrm{rank}(\boldsymbol{Z}_l^T)$ is at most $b$, $\mathrm{rank}(\frac{\partial \mathcal{L}}{\partial \boldsymbol{W}_l^Q}) = \min(\mathrm{rank}(\boldsymbol{Z}_l^T), \mathrm{rank}(\frac{\partial \mathcal{L}}{\partial \boldsymbol{Q}_l})) = \mathrm{rank}(\boldsymbol{Z}_l^T)$. This finishes the proof.  □

**Theorem 5.2.** *When $b < d$, the probability of a random vector $\in \mathbb{R}^d$ to be part of $\mathrm{colspan}(\frac{\partial \mathcal{L}}{\partial \boldsymbol{W}_l^Q})$ is almost surely $0$.*

*Proof.* As $\frac{\partial \mathcal{L}}{\partial \boldsymbol{W}_l^Q}$ is rank-deficient (Theorem 3.1), $\mathrm{colspan}(\frac{\partial \mathcal{L}}{\partial \boldsymbol{W}_l^Q})$ is a strict linear subspace of $\mathbb{R}^d$. Thus, it has a hypervolume $0$ via the Sard's lemma. This directly implies that the probability of a random vector $\in \mathbb{R}^d$ to be part of $\mathrm{colspan}(\frac{\partial \mathcal{L}}{\partial \boldsymbol{W}_l^Q})$ is almost surely $0$.  □

### B.2  Technical Assumptions of DAGER

Below we present a brief commentary on the technical assumptions of DAGER. DAGER makes three assumptions:

- We assume that $\frac{\partial \mathcal{L}}{\partial \boldsymbol{Q}_l}$ is full-rank.
- We require the total number of tokens $b$ in the batch to be smaller than the embedding dimension $d$, ensuring that $\frac{\partial \mathcal{L}}{\partial \boldsymbol{W}_l^Q}$ is of low-rank.
- We assume a known discrete set of possible inputs to the model, i.e. its vocabulary.

Importantly, DAGER does not assume any prior knowledge of the labels or the lengths of each sequence in the batch nor access to the gradients of the embedding layers which have been shown to leak significant information [26]. Further, we require no language priors and operate under the honest-but-curious setting which does not allow malicious changes to model weights. Finally, sub-differentiability is sufficient for applying DAGER.

In practice, DAGER requires much fewer assumptions than existing works in the honest-but-curious setting while being successful in a variety of common LLM tasks, e.g., next-token prediction and sentence classification. While these tasks use the cross-entropy loss, DAGER can be applied to any loss function that non-trivially depends on every input token. This ensures the full-rankness of $\frac{\partial \mathcal{L}}{\partial \boldsymbol{W}_l^Q}$.

To confirm its generality, we apply DAGER with a Frobenius norm-based loss and ReLU activation functions. We use a custom loss function $\mathcal{L}(s_1, s_2, \ldots, s_P) = \|\boldsymbol{f}^L(s_1, s_2, \ldots, s_P)\|_F$, where $\|.\|_F$ is the Frobenius norm, which is equivalent to an MSE loss with $\boldsymbol{0}$ as a target vector. We report the results of applying DAGER with these modifications on Rotten Tomatoes using GPT-2 with $B = 16$ in Table 11, achieving ROUGE-1 and ROUGE-2 scores of >99% in both cases.

### B.3   DAGER under the FedAvg protocol

We establish that DAGER can be effectively used to attack clients employing FedAvg updates under mild assumptions. First, we show that it is possible to theoretically apply Theorem 5.1 to the protocol directly on the first layer, and under reasonably low model updates for further layers. We then demonstrate experimentally that DAGER can be successfully applied across a wide range of parameters, namely the number of epochs $E$, learning rate $\eta$ and mini-batch size $b_{mini}$.

**Spanchecks for FedAvg updates**   Let $\boldsymbol{X}_l^e = f^l(s_1^e, s_2^e, \ldots, s_{b_{mini}}^e)$ denote the input embedding vectors to the relevant $l$-th layer query projection matrix for a mini-batch of size $b_{mini}$. Obtaining $\boldsymbol{X}_l^e$ in the forward pass is equivalent to sampling the corresponding rows in the full-batch representation $\boldsymbol{X}_l = f^l(s_1^e, s_2^e, \ldots, s_B)$, as each sequence in a batch is independent from all the rest. Therefore, we can rewrite each local gradient update $\frac{\partial \mathcal{L}}{\partial \boldsymbol{W}_l^Q} = \boldsymbol{X}_l^{eT} \frac{\partial \mathcal{L}}{\partial \boldsymbol{Q}_l^e}$ as $\boldsymbol{X}_l^T \frac{\partial \mathcal{L}}{\partial \boldsymbol{Q}_l}$, where the $i$-th column of $\frac{\partial \mathcal{L}}{\partial \boldsymbol{Q}_l}$ is the zero vector $\boldsymbol{0}$ if $s_i$ is not present in the mini-batch. Therefore, we are able to simply disregard the mini-batch sampling and focus only on the total number of iterations $\mathcal{E}$, assuming that every sequence in the original batch is sampled at least once.

Let the input embedding vectors to the $l$-th layer at timestep $t < \mathcal{E}$ be $\boldsymbol{X}_l^t$. The final weight $\boldsymbol{W}_l^{\mathcal{E}}$ after $\mathcal{E}$ steps can be written as:

$$\boldsymbol{W}_l^{\mathcal{E}} = \boldsymbol{W}_l^0 - \eta \sum_{t=0}^{\mathcal{E}} \frac{\partial \mathcal{L}}{\partial \boldsymbol{W}_l^t} = \boldsymbol{W}_l^0 - \eta \sum_{t=0}^{\mathcal{E}} \boldsymbol{X}_l^{t^T} \frac{\partial \mathcal{L}}{\partial \boldsymbol{Q}_l^t} \tag{4}$$

Under the assumption that the changes in model weights are relatively small, we can approximate $\boldsymbol{X}_l^t = \boldsymbol{X}_l^0$. This is always the case for $l = 0$, as the embeddings before the first layer are independent of the model weights. This allows us to rewrite $\boldsymbol{W}_{\mathcal{E}}$ as:

$$\boldsymbol{W}_l^{\mathcal{E}} = \boldsymbol{W}_l^0 - \eta \sum_{t=0}^{\mathcal{E}} \boldsymbol{X}_l^{t^T} \frac{\partial \mathcal{L}}{\partial \boldsymbol{Q}_l^t} = \boldsymbol{W}_l^0 - \eta \sum_{t=0}^{\mathcal{E}} \boldsymbol{X}_l^{0^T} \frac{\partial \mathcal{L}}{\partial \boldsymbol{Q}_l^t} = \boldsymbol{W}_l^0 - \boldsymbol{X}_l^{0^T} \left( \eta \sum_{t=0}^{\mathcal{E}} \frac{\partial \mathcal{L}}{\partial \boldsymbol{Q}_l^t} \right) \tag{5}$$

As the server has knowledge of the starting weight $\boldsymbol{W}_l^0$ and the final weight $\boldsymbol{W}_l^{\mathcal{E}}$, it is able to compute the sum of all gradient steps, i.e we will be able to apply Theorem 5.1 to $\boldsymbol{X}_l^{0^T} (\eta \sum_{t=0}^{\mathcal{E}} \frac{\partial \mathcal{L}}{\partial \boldsymbol{Q}_l^t})$. We still require $\sum_{t=0}^{\mathcal{E}} \frac{\partial \mathcal{L}}{\partial \boldsymbol{Q}_l^t}$ to be full-rank, which is satisfied under standard DAGER assumptions.

**Further experimental details**   We further empircally demonstrate that DAGER can effectively utilise the assumption of consistent feature embeddings across epochs under a reasonable learning rate and number of iterations. As shown in Table 4, we observe near-exact reconstruction rates for most configurations, with metrics only slightly declining as the number of epochs increases. The key factor that influences the success of DAGER is observed to be the learning rate $\eta$, as the aforementioned

assumption might be invalidated at large $\eta$. However, learning rates exceeding $\eta \geq 10^{-3}$ are typically too high for the model to converge, particularly in multi-client settings. Therefore, we can conclude that DAGER is highly effective in the FedAvg context.

## B.4 DAGER under LoRA training

In this section, we discuss how DAGER can be extended to work on LoRA weight decomposition. Under LoRA, the linear layer weight updates for a weight $\boldsymbol{W} \in \mathbb{R}^{d \times d}$ are performed on a low-rank representation: $\boldsymbol{W} = \boldsymbol{W}_0 + \boldsymbol{A}\boldsymbol{B}$, where $\boldsymbol{A} \in \mathbb{R}^{d \times r}, \boldsymbol{B} \in \mathbb{R}^{r \times d}$. As we obtain the gradient weights for both $\boldsymbol{A}$ and $\boldsymbol{B}$, we can apply Theorem 5.1 to $\boldsymbol{A}$ with $\boldsymbol{Z}_A = \boldsymbol{X}\boldsymbol{A}$, namely because $\frac{\partial \mathcal{L}}{\partial \boldsymbol{A}} = \boldsymbol{X}^T \frac{\partial \mathcal{L}}{\partial \boldsymbol{Z}_A}$. Assuming that $\frac{\partial \mathcal{L}}{\partial \boldsymbol{X}\boldsymbol{A}}$ is full-rank and that $b < r$, our work is directly applicable. This can replace the spanchecks to be performed on $\boldsymbol{A}$ instead of $\boldsymbol{W}$ for each layer, after which DAGER can be applied directly. In practice, LoRA finetuning typically initializes $\boldsymbol{W} = \boldsymbol{W}_0$ and $\boldsymbol{B}$ to only contain zeroes which reduce the rank of $\frac{\partial \mathcal{L}}{\partial \boldsymbol{A}}$ for the first few optimization steps. We therefore train the LLaMa-3.1 8B model on the Rotten Tomatoes dataset using a batch size of 4 with $r = 256$ (following [39]) for 3 epochs before applying DAGER. We report results in Table 11 and observe an excellent R1 and R2 of about 95%.

## B.5 Complexity Analysis

A key point of DAGER is the algorithm's exceptional computational efficiency on decoder-based models. In order to quantify the dependency of runtime on relevant variables, we describe the asymptotic complexity for both decoder- and encoder-based models. Below we list and prove several lemmas that assist us in the complete proof of our assertion for the complexity of DAGER. When not specified, a batch size of $B = 1$ is implied for any inputs.

**Lemma B.1.** *The product of two matrices $\boldsymbol{M}^1 \in \mathbb{R}^{n \times m}$ and $\boldsymbol{M}^2 \in \mathbb{R}^{m \times p}$ can be naively computed in $\mathcal{O}(nmp)$.*

*Proof.* We write down the product:

$$(\boldsymbol{M}^1 \boldsymbol{M}^2)_{ij} = \sum_k \boldsymbol{M}^1_{ik} \boldsymbol{M}^2_{kj}$$

To produce the entire matrix we explore all integers $i = 1 \ldots n$, $j = 1 \ldots p$ and $k = 1 \ldots m$, and in particular any combination of the 3. This implies that we make $nmp$ iterations, from which a time complexity of $\mathcal{O}(nmp)$ follows. □

**Lemma B.2.** *For any matrix $\boldsymbol{M} \in \mathbb{R}^{d \times n}$ and vector $\boldsymbol{v} \in \mathbb{R}^d$, we can compute the projection of $\boldsymbol{v}$ on the subspace spanned by the columns of $\boldsymbol{M}$ in $\mathcal{O}(d^2 n)$ time.*

*Proof.* Projecting onto the column space of a matrix can be done by projecting the vector onto individual columns and then summing all projected components. We compute this using Einstein notation, while denoting the resulting vector as $\boldsymbol{p} \in \mathbb{R}^d$. Then, we obtain:

$$\boldsymbol{p}_k = \sum_{i,j} \boldsymbol{M}_{ki} \boldsymbol{M}_{ji} \boldsymbol{v}_j$$

This loop iterates over $i = 1 \ldots n, j = 1 \ldots d, k = 1 \ldots d$, resulting in a total number of $d^2 n$ iterations, which implies a time complexity of $\mathcal{O}(d^2 n)$. □

**Lemma B.3.** *For any transformer-based model, which has an embedding dimension $d$, square projection weights of dimension $d \times d$, and an MLP hidden dimension of $d_{MLP}$, propagating a sequence of $b$ tokens, takes time of asymptotic complexity $\mathcal{O}(bd^2 + b^2 d + bd_{MLP}d)$.*

*Proof.* We follow the notation defined in Sec. 3.1. The embedding representation of the sequence is denoted as $\boldsymbol{z} \in \mathbb{R}^{b \times d}$. We then obtain the query, key and value vectors $\boldsymbol{Q}, \boldsymbol{K}, \boldsymbol{V}$ by multiplying with the weight matrices $\boldsymbol{W}_1^Q, \boldsymbol{W}_1^K, \boldsymbol{W}_1^V \in \mathbb{R}^{d \times d}$. According to Lemma B.1, we can accomplish this in a time of $\mathcal{O}(bd^2)$.

Computing the attention scores is dominated by computing $\boldsymbol{QK}^T$, which by applying Lemma B.1, can be done in $\mathcal{O}(b^2d)$.

As a final step to the self-attention component, we compute the multiplication of the scores $A = \text{softmax}(\boldsymbol{M} \odot \frac{\boldsymbol{QK}^T}{\sqrt{d}}) \in \mathbb{R}^{b \times b}$ and $\boldsymbol{V} \in \mathbb{R}^{b \times d}$ in time $\mathcal{O}(b^2d)$. We note that computing the row-wise softmax operations can be amortized and take a total of $\mathcal{O}(b^2)$ time, which is dominated by $\mathcal{O}(b^2d)$. The total computation up until this point can, therefore, be done in $\mathcal{O}(bd^2 + b^2d)$.

To produce the output of the transformer block, we pass the self-attention through an MLP layer, which can be represented by a set of simple matrix multiplications (of the same complexity). Therefore, applying Lemma B.1 once again, this step requires time $\mathcal{O}(bd_{\text{MLP}}d)$.

We can sum up all components to obtain a total complexity of $\mathcal{O}(bd^2 + b^2d + bd_{\text{MLP}}d)$. If we reasonably assume that $d_{\text{MLP}} = \mathcal{O}(d)$, then the complexity can be simplified to $\mathcal{O}(bd^2 + b^2d)$. $\quad\square$

Having explored in-depth each smaller component of the algorithm, we can now determine the complexity of DAGER. We explore 3 instances of our algorithm, namely separating decoder- and encoder-based models, while also differentiating between absolute positional encodings and RoPE. We summarize our findings in a single theorem:

**Theorem B.4.** *By considering a training iteration of batch of size $B$ with the longest sentence being of length $P$, where the sentences are represented as a sequence of tokens from a vocabulary of size $V$, DAGER can reconstruct the input with an asymptotic time complexity of:*

1. *For decoder-based models:*

   (a) *If the model applies positional embeddings before layer 0 - $\mathcal{O}(P^2BVd^2 + d^3 + P^3B^3d^2)$*

   (b) *If the model applies positional embeddings after the first projection, i.e. RoPE - $\mathcal{O}(PBVd^2 + d^3 + P^4B^3d^2)$*

2. *For encoder-only models - $\mathcal{O}(P^2BVd^2 + d^3 + B^PP^2d^2)$*

*Proof.* We separate our algorithm in 2 different parts:

1. Recovering the tokens $\mathcal{T}^*$ through the span check on $\boldsymbol{W}_1^Q$.

2. Reconstructing the entire sequences $\mathcal{S}^*$.

**Token recovery**   We begin by describing the first step - recovering individual tokens per position. We begin by obtaining $\mathcal{T}^*$ by taking $\mathcal{T}_i^* = \{v \in \mathcal{V} | f^0(v, i) \in \text{colspan}(\frac{\partial \mathcal{L}}{\partial \boldsymbol{W}_1^Q})\}$. Because we determine the span check via the projection distance $d_{\text{proj}}$ of a representation vector $\boldsymbol{z} \in \mathbb{R}^d$, and truncated gradient $\frac{\partial \mathcal{L}}{\partial \boldsymbol{W}_l^Q} \in \mathbb{R}^{d \times r}$, where $r = \text{rank}(\frac{\partial \mathcal{L}}{\partial \boldsymbol{W}_1^Q})$ according to Lemma B.2, this can be done in time $\mathcal{O}(d^2r)$. The rank of both $\frac{\partial \mathcal{L}}{\partial \boldsymbol{W}_1^Q}, \frac{\partial \mathcal{L}}{\partial \boldsymbol{W}_2^Q}$ is limited by the total number of tokens $b \in \mathcal{O}(PB)$. As can be inferred by Theorem 5.1, the complexity of the spancheck is $\mathcal{O}(PBd^2)$. We perform this for every token in the vocabulary for an additional factor of $V$, making the total complexity $\mathcal{O}(PBVd^2)$. We note that for models which employ positional embeddings before the first transformer block, we repeat this step $P$ times, while we only have to do it once for those that don't. This respectively results in time complexities of $\mathcal{O}(P^2BVd^2)$ and $\mathcal{O}(PBVd^2)$ for recovering individual tokens.

**Spancheck complexity**   In practice, we perform all span checks by performing a Singular Value Decomposition on $\frac{\partial \mathcal{L}}{\partial \boldsymbol{W}_1^Q}$ and then applying the projection distance to the right orthonormal. It is a well known result that SVD for a matrix of size $d \times d$ takes time $\mathcal{O}(d^3)$.

**Sequence reconstruction**   Having recovered the individual tokens, we detail the resulting time complexity of reconstructing the entire sequences. We describe each step for decoder-based models, before proceeding to encoder-based ones.

- For decoder-based models we begin by *describing the number of tokens we obtain per position*. If the model applies positional embedding before the first transformer layer, we assume that for each position $i$ we recover a set of tokens $\mathcal{T}_i^*$, such that $|\mathcal{T}_i^*| = \mathcal{O}(T)$ for some variable $T$ that depends on our input parameters and setup. Similarly, for models that do not have positional embeddings before the first transformer layer, we only recover a single set of tokens $\mathcal{T}^*$ of size $|\mathcal{T}^*| = \mathcal{O}(PT)$.

- We now show the complexity required for performing the *forward pass and span check for a single sequence*. For a sequence length $n$, according to Lemma B.3 the forward pass takes time $\mathcal{O}(nd^2 + n^2 d)$, and the span check per token is of complexity $\mathcal{O}(PBd^2)$ (as per Lemma B.2). This implies a time complexity of $\mathcal{O}(nPBd^2)$ per sequence for the span check, leading us to an overall complexity of $\mathcal{O}(nPBd^2 + nd^2 + n^2 d)$.

- Finally, we apply our observations to the *full reconstruction*. We further assume that at each step of the greedy reconstruction, we maintain $\mathcal{O}(B)$ possible sequences, which is usually the case when performing a greedy reconstruction. By repeating the above propagation for every combination per sequence length yields a total runtime of $\mathcal{O}(T \times B \times (nd^2 + n^2 d + nPBd^2)) = \mathcal{O}(PTB^2d^2 n)$ and $\mathcal{O}(PT \times B \times (nd^2 + n^2 d + nPBd^2)) = \mathcal{O}(P^2TB^2d^2 n)$ in the cases described in statement 1a) and 1b) respectively.

- *Summing over all lengths* yields a time of $\mathcal{O}(\sum_{n=1}^P PB^3d^2 n) = \mathcal{O}(P^3TB^2d^2)$ for the former. Analogically, for the latter we obtain a complexity of $\mathcal{O}(P^4TB^2d^2)$.

- In practice, we observe that $T = \alpha B$ for some factor $\alpha$ between 1 and 10, hence we can assume that $T = \mathcal{O}(B)$. This is a practically correct assumption for a reasonably defined threshold $\tau_1$. This makes the final time complexity for recovering sequences $\mathcal{O}(P^3TB^2d^2) = \mathcal{O}(P^3B^3d^2)$ for case 1a) and $\mathcal{O}(P^4TB^2d^2) = \mathcal{O}(P^4B^3d^2)$ for 1b).

On the other hand, in the case of encoder-based models, we need to exhaust all possible token combinations over all positions.

- We again assume that for each position $i$ we recover a set of tokens $\mathcal{T}_i^*$, such that $|\mathcal{T}_i^*| = \mathcal{O}(T)$.

- Because we have to explore all possible combinations, that results in a total number of $\prod_{i=1}^P |\mathcal{T}_i^*| = \prod_{i=1}^P \mathcal{O}(T) = \mathcal{O}(T^P)$ sequences.

- We now show the cost of the span check on a single sequence. We leverage our finding that one such span check takes time $\mathcal{O}(PBd^2)$ (we again substitute that the rank $r = \mathcal{O}(PB)$ and apply Lemma B.2), meaning all span checks take time $\mathcal{O}(P^2Bd^2)$ per sequence.

- Additionally, the forward pass takes $\mathcal{O}(Pd^2 + P^2d)$ time.

- Finally, because we reconstruct each sequence separately, we repeat the reconstruction algorithm $\mathcal{O}(B)$ times. This leads to the complexity for this step - $\mathcal{O}(T^P B(P^2Bd^2 + Pd^2 + P^2d)) = \mathcal{O}(T^P B^2 P^2 d^2)$.

- As above, we assume $T = \mathcal{O}(B)$, leading us to a *final complexity* of $\mathcal{O}(B^P B^2 P^2 d^2) = \mathcal{O}(B^P P^2 d^2)$.

Combining our conclusions for steps 1 and 2 yields the stated complexities for each setting. □

The key points to highlight is that DAGER is polynomial across both sequence length and batch size for decoders, regardless of the type of positional encoding. Meanwhile, for encoders the we observe that DAGER is exponential in length and polynomial with high degree in terms of batch size. To this end, the heuristics we described are crucial to significantly reduce the search space.

## B.6 Encoder Algorithm

Table 6: Specifications of models that were used in our work.

| Model | Type | No. layers | $d$ | No. heads | Feed-forward size | $V$ | Positional embedding | No. Parameters |
|---|---|---|---|---|---|---|---|---|
| GPT-2$_{\text{BASE}}$ | Decoder | 12 | 768 | 12 | 3072 | 50,257 | Absolute | 137M |
| GPT-2$_{\text{LARGE}}$ | Decoder | 36 | 1280 | 20 | 5,120 | 50,257 | Absolute | 812M |
| LLaMa-2 (7B) | Decoder | 32 | 4,096 | 32 | 11,008 | 32,000 | RoPE | 6.74B |
| LLaMa-3.1 (8B) | Decoder | 32 | 4,096 | 32 | 14,336 | 128,256 | RoPE | 8.03B |
| LLaMa-3.1 (70B) | Decoder | 80 | 8,192 | 64 | 28,672 | 128,256 | RoPE | 70.6B |
| BERT$_{\text{BASE}}$ | Encoder | 12 | 768 | 12 | 3072 | 30,522 | Absolute | 110M |

In this section, we provide pseudocode for DAGER when applied to encoder-based LLMs. We provide it in Algorithm 3, where we first find the set of client sequence lengths $n_j$ and store them in $\mathcal{N}$ (Line 4 to Line 8). We then go through the $n_j$s from the smallest to the largest (Line 9), enumerating all possible sequences $s \in \mathcal{S}_i$ of length $i$ (Line 10). Importantly, when a correct sentence $s \in \mathcal{S}_i^{\text{corr}}$ is found its tokens are removed for the token sets $\mathcal{T}_i^*$ (Line 14). Finally, we return the deduplicated best reconstructions across different sequence lengths $\mathcal{S}_{\text{best}}^*$ (Line 15). We note that when $\mathcal{S}_i$ is larger than 10M combinations, we sample 10M random combinations from it instead.

**Algorithm 3** DAGER for Encoders

1: **function** ATTENC($T, B, \frac{\partial \mathcal{L}}{\partial \boldsymbol{W}_1^Q}, \frac{\partial \mathcal{L}}{\partial \boldsymbol{W}_2^Q}, V, P, f^{0/1}, \tau_{1/2}$)
2: $\quad n, \mathcal{T}^*, \mathcal{D}^* \leftarrow$ GETTOK($\frac{\partial \mathcal{L}}{\partial \boldsymbol{W}_1^Q}, V, P, f^0, \tau_1$)
3: $\quad \mathcal{T}^* \leftarrow$ TOPBTOKENS($\mathcal{T}^*, \mathcal{D}^*, B$)
4: $\quad \mathcal{N} \leftarrow \{\}$
5: $\quad$ **for** $i \leftarrow 1, \dots, n$ **do**
6: $\quad\quad$ **if** EOS $\in \mathcal{T}_i^*$ **then**
7: $\quad\quad\quad \mathcal{N} \leftarrow \mathcal{N} + \{i+1\}$
8: $\quad\quad\quad \mathcal{T}_i^* \leftarrow \mathcal{T}_i^* \setminus \{\text{EOS}\}$
9: $\quad$ **for** $i \in$ SORT($\mathcal{N}$) **do**
10: $\quad\quad \mathcal{S}_i \leftarrow \mathcal{T}_1^* \times \cdots \times \mathcal{T}_i^*$
11: $\quad\quad \mathcal{S}_i^{\text{corr}} \leftarrow \{s \in \mathcal{S}_i \mid d(f_p^1(s), 2) < \tau_2. \forall p \in [i]\}$
12: $\quad\quad$ **for** $s \in \mathcal{S}_i^{\text{corr}}$ **do**
13: $\quad\quad\quad$ **for** $p \leftarrow 1, \dots, i$ **do**
14: $\quad\quad\quad\quad \mathcal{T}_p^* \leftarrow \mathcal{T}_p^* \setminus \{s_p\}$
15: $\quad S_{\text{best}}^* \leftarrow$ TOPUNIQUE($\bigcup_{i=1}^l S_i, \frac{\partial \mathcal{L}}{\partial \boldsymbol{W}_2^Q}, B$)
16: $\quad$ **return** $S_{\text{best}}^*$

## C  Additional experimental details

In this section we describe additional details regarding the experimental setup for DAGER.

**Models**  In particular, we explore its feasibility on the base variant GPT-2$_{\text{BASE}}$, featuring a 12-layer transformer with a 768-dimensional hidden state, 12 attention heads and a feed-forward filter of size 3072. Additionally, we examine the effects of model size by considering GPT-2$_{\text{LARGE}}$, which contains 36 layers, a 1280-dimensional hidden state, 20 attention heads, and a feed-forward filter of size 5120. To demonstrate the versatility of our approach, we also evaluate our attack in a next-token prediction context on the GPT-2$_{\text{BASE}}$ model. We further demonstrate the efficacy of our attack on LLaMa 2-7B[13] and BERT$_{\text{BASE}}$[14] to illustrate our performance on a state-of-the-art decoder and an encoder-only model, respectively. All models were sourced from HuggingFace[41] in their pre-trained form, following standard practice in language modeling research[42]. Full specification details for all models can be found in Table 6.

**Datasets**  In our evaluation, we utilize three binary classification datasets with varying sentence lengths, namely the Corpus of Linguistic Acceptability (**CoLA**)[15], the Stanford Sentiment Treebank (**SST-2**)[16], which are part of the **GLUE** benchmark[43], as well as the **Rotten Tomatoes**[17] sentiment analysis dataset. While our algorithm is data-independent, previous studies have indicated that text size can affect reconstructability[11, 6]. We demonstrate robustness to this factor by selecting the aforementioned datasets, with CoLA featuring text typically ranging from 4 to 9 words, SST-2 - from 4 to 13 words, and Rotten Tomatoes - from 10 to 27 words. We note that the binary classification setting has been shown to be a less vulnerable than next-token prediction which can be attacked via label reconstruction attacks [35], however, we conduct an additional experiment to substantiate the claim in this setting. Finally, to showcase DAGER's capability to handle arbitrarily long sequences, we leverage the **European Court of Human Rights (ECHR)**[18] dataset which includes sentences that are *over 1000 words long*, far exceeding the maximum input length of any of the aforementioned models.

Table 7: Total runtime for all main experiments given in hours. Fields containing N/A represent experiments that were not run.

| | | | B=1 | B=2 | B=4 | B=8 | B=16 | B=32 | B=64 | B=128 |
|---|---|---|---|---|---|---|---|---|---|---|
| GPT-2 | CoLA | TAG | 9.4 | 9.1 | 9.2 | 9.5 | *N/A* | *N/A* | *N/A* | *N/A* |
| | | LAMP | 15.9 | 24.2 | 40.7 | 68.3 | *N/A* | *N/A* | *N/A* | *N/A* |
| | | DAGER | 1.5 | 1.1 | 1.5 | 2.8 | 8.0 | 8.5 | 4.4 | 8.2 |
| | SST-2 | TAG | 9.5 | 9.7 | 9.1 | 9.7 | *N/A* | *N/A* | *N/A* | *N/A* |
| | | LAMP | 16.3 | 24.2 | 39.0 | 66.5 | *N/A* | *N/A* | *N/A* | *N/A* |
| | | DAGER | 1.1 | 1.4 | 1.7 | 3.6 | 8.8 | 13.7 | 8.3 | 7.1 |
| | Rotten Tomatoes | TAG | 8.6 | 8.8 | 9.2 | 9.7 | *N/A* | *N/A* | *N/A* | *N/A* |
| | | LAMP | 16.2 | 24.2 | 37.9 | 67.4 | *N/A* | *N/A* | *N/A* | *N/A* |
| | | DAGER | 1.8 | 1.7 | 2.5 | 3.5 | 8.5 | 10.2 | 1.4 | 1.9 |
| BERT | CoLA | TAG | 6.1 | 6.3 | 9.2 | 11.6 | *N/A* | *N/A* | *N/A* | *N/A* |
| | | LAMP | 11.0 | 26.4 | 42.6 | 85.2 | *N/A* | *N/A* | *N/A* | *N/A* |
| | | DAGER | 0.1 | 0.1 | 1.3 | 19.4 | *N/A* | *N/A* | *N/A* | *N/A* |
| | SST-2 | TAG | 9.1 | 6.5 | 7.9 | 11.8 | *N/A* | *N/A* | *N/A* | *N/A* |
| | | LAMP | 17.1 | 25.8 | 43.8 | 82.8 | *N/A* | *N/A* | *N/A* | *N/A* |
| | | DAGER | 0.1 | 0.7 | 11.8 | 59.1 | *N/A* | *N/A* | *N/A* | *N/A* |
| | Rotten Tomatoes | TAG | 8.5 | 8.6 | 8.6 | 11.2 | *N/A* | *N/A* | *N/A* | *N/A* |
| | | LAMP | 17.4 | 28.2 | 29.8 | 83.1 | *N/A* | *N/A* | *N/A* | *N/A* |
| | | DAGER | 0.1 | 7.6 | 48.9 | 195.0 | *N/A* | *N/A* | *N/A* | *N/A* |
| LLaMA-2 | CoLA | DAGER | 7.7 | 8.6 | 8.2 | 9.9 | 12.6 | 21.2 | 41.2 | 160.0 |
| | SST-2 | DAGER | 7.8 | 7.7 | 10.0 | 12.5 | 19.1 | 45.9 | 74.3 | 257.7 |
| | RT | DAGER | 7.6 | 9.1 | 10.7 | 15.6 | 26.0 | 39.5 | 112.2 | 523.1 |

**Computational requirements**   We implement DAGER in PyTorch [44] and run all experiments on a single GPU. Tests on the LLaMa-2 (7B) architecture were performed on NVIDIA A100 Tensor Core GPUs, which boast 40 GB of memory, while all others were ran on NVIDIA L4 GPUs with 24 GB of memory. In practice, less demanding resources may be used, especially for lower batch sizes on BERT and GPT-2$_{\text{BASE}}$. In terms of required RAM, we used between 16 GB and 150 GB per experiment, depending on the batch size and model.

**Hyperparameter details**   We use a span check acceptance threshold of $\tau_1 = 10^{-5}$ in the first layer, and $\tau_2 = 10^{-3}$ in the second, a rank truncation of $\Delta_b = 20$, and for decoder-based models consider at most $10\,000\,000$ proposal sentences per recovered EOS token position. We consider pre-trained models with a randomly initialized classification head using a normal distribution with $\sigma = 10^{-3}$. To manage numerical instabilities within the framework, we tweak the eigenvalue threshold when doing the SVD $\tau_l^{\text{rank}}$ and decrease with the batch size growing, varying it between $10^{-7}$ and $10^{-9}$.

## C.1   Runtime of experiments

We further demonstrate the computational efficiency of DAGER. The runtime summary can be found in Table 7. It is notable that for the experiments on BERT, we have a drastic increase in complexity with respect to the batch size and sequence length, as expected from App. B.5, which do not affect the baselines as much. That said, we still remain within the same order of magnitude, while achieving significantly better results.

In contrast, we achieve a significant improvement for decoder-based models. We notice that within the attack's scope for GPT-2, we can reconstruct a batch for less than *8 minutes* for any batch size. It is important to highlight that the runtime decreases for the batch sizes of 64 and 128 because we are more often than not unable to recover any tokens due to the embedding dimension limitation described in Sec. 7.

Table 8: Ablation study on GPT-2$_{\text{BASE}}$ with and without the rank threshold heuristic. We report scores of 0 for any example that has full-rank gradients. R-1 and R-2 denote the ROUGE-1 and ROUGE-2 scores respectively.

| | | $B = 16$ | | $B = 32$ | | $B = 64$ | | $B = 128$ | |
| | | R-1 | R-2 | R-1 | R-2 | R-1 | R-2 | R-1 | R-2 |
|---|---|---|---|---|---|---|---|---|---|
| CoLA | With cutoff | $100.0 \pm 0.0$ | $100.0 \pm 0.0$ | $100.0 \pm 0.0$ | $100.0 \pm 0.0$ | $100.0 \pm 0.0$ | $100.0 \pm 0.0$ | $30.3 \pm 1.0$ | $14.6 \pm 0.9$ |
| | No cutoff | $100.0 \pm 0.0$ | $100.0 \pm 0.0$ | $100.0 \pm 0.0$ | $100.0 \pm 0.0$ | $100.0 \pm 0.0$ | $100.0 \pm 0.0$ | $0.0 \pm 0.0$ | $0.0 \pm 0.0$ |
| SST-2 | With cutoff | $100.0 \pm 0.0$ | $94.6 \pm 1.1$ | $100.0^{+0.0}_{-0.1}$ | $93.4 \pm 1.0$ | $92.9 \pm 3.0$ | $85.0 \pm 3.5$ | $13.7 \pm 1.4$ | $4.3 \pm 0.5$ |
| | No cutoff | $100.0 \pm 0.0$ | $94.6 \pm 1.1$ | $100.0^{+0.0}_{-0.1}$ | $93.4 \pm 1.0$ | $75.0 \pm 8.7$ | $69.7 \pm 8.2$ | $0.0 \pm 0.0$ | $0.0 \pm 0.0$ |
| Rotten Tomatoes | With cutoff | $100.0 \pm 0.0$ | $99.9^{+0.1}_{-0.3}$ | $98.0 \pm 1.7$ | $97.8 \pm 1.8$ | $2.8 \pm 1.1$ | $1.1 \pm 0.4$ | $0.0 \pm 0.0$ | $0.0 \pm 0.0$ |
| | No cutoff | $100.0 \pm 0.0$ | $99.9^{+0.1}_{-0.3}$ | $93.9 \pm 4.8$ | $93.8 \pm 4.8$ | $0.0 \pm 0.0$ | $0.0 \pm 0.0$ | $0.0 \pm 0.0$ | $0.0 \pm 0.0$ |

Table 9: Example comparison between DAGER and LAMP$_{\text{Cos}}$ for random samples from different datasets, reconstructed at batch size 1. We note that LAMP$_{\text{Cos}}$ was selected, as it was the best-performing baseline model for a batch size of 1.

| | | Sequence |
|---|---|---|
| CoLA | Reference | Sarah devoured the cakes in the kitchen last night. |
| | DAGER | Sarah devoured the cakes in the kitchen last night. |
| | LAMP$_{\text{Cos}}$ | Sarah imaginary even kitchen dev devoured cakes last night |
| SST-2 | Reference | a caper that's neither original nor terribly funny |
| | DAGER | a caper that's neither original nor terribly funny |
| | LAMP$_{\text{Cos}}$ | a that's neither an perennRe nor terribly funny |
| Rotten Tomatoes | Reference | plays like the old disease-of-the-week small-screen melodramas. |
| | DAGER | plays like the old disease-of-the-week small-screen melodramas. |
| | LAMP$_{\text{Cos}}$ | plays it like the old screen impactnorm disease . small-screen like melodramas. |

## C.2 Rank restriction ablation study

In section Sec. 5.3, we described that for full-rank matrices we attempt a best-effort reconstruction by artificially restricting the rank by a threshold $\tilde{b}$. Here we demonstrate the effect of this component by showing the performance of DAGER without attempting to recover any part of the sentence. Any time we observe a sample with full-rank gradients at either the first or second layers, we immediately fail and report a score of 0. The results can be seen in Table 8.

## C.3 Example reconstructions

In Table 9 we show sample reconstructions between the best-performing baseline LAMP_Cos for a batch size $B = 1$. We note that DAGER achieves exact reconstruction, while LAMP_Cos can predict less than half of the sentence correctly.

## C.4 DAGER under differential privacy

In this section, we show how DAGER performs under a defended setting, in particular by adding random Gaussian noise with variance $\sigma^2$ to all gradients. We explore the range $\sigma \in [10^{-5}, 5 \times 10^{-4}]$, as for any $\sigma \geq 10^{-3}$, the sentiment prediction accuracy of the converged model drops to below 80% from $> 87\%$. We apply DAGER on the GPT-2 model for the Rotten Tomatoes dataset at $B = 1$. Due to the highly random nature of this type of defense, we cannot simply filter the sequences by measuring the single span check distance at layer $l = 2$. Instead, we utilise that further layers $l > 3$ retain the same property that the input embeddings only depend on previous tokens, and measure the average $\bar{d} = \sum_{l=2}^{L_{DP}} d(f_i^{l-1}(s), l)$ for a certain number of layers $L_{DP}$, which we optimise as a hyperparameter. Further, because any noise will make the gradient updates full-rank, for computation purposes we set a constant rank of $r = 100$, which is much higher than the length of any sentence. While it has been shown [45] that differencial privacy provides provable guarantees for protecting privacy, we provide promising initial results, as seen in Table 10. We believe that there are numerous improvements one could make, but leave these for future work.

Table 10: Experiments on the differential privacy setting under Gaussian noise on the GPT-2 model with a batch size of 1 on the Rotten Tomatoes dataset. R-1 and R-2 denote the ROUGE-1 and ROUGE-2 scores respectively.

| $\sigma = 10^{-5}$ | | $\sigma = 5 \times 10^{-5}$ | | $\sigma = 10^{-4}$ | | $\sigma = 5 \times 10^{-4}$ | |
|---|---|---|---|---|---|---|---|
| R-1 | R-2 | R-1 | R-2 | R-1 | R-2 | R-1 | R-2 |
| $74.0 \pm 3.2$ | $70.8 \pm 3.4$ | $46.9 \pm 4.1$ | $32.5 \pm 4.0$ | $20.7 \pm 4.4$ | $11.6 \pm 3.4$ | $5.6^{+2.6}_{-1.9}$ | $0.9^{+3.3}_{-0.7}$ |

Table 11: Miscallaneous experiments, referenced in the evaluation section. We applied DAGER on the Rotten Tomatoes dataset for $B = 16$, if not specified otherwise.

| | LLaMa-3 70B ($B = 1$) | LLaMa-3.1 8B | Frobenius norm loss | ReLU activation | LoRA ($r = 256$) |
|---|---|---|---|---|---|
| R-1 | $99.9^{+0.1}_{-0.2}$ | $99.4^{+0.1}_{-0.3}$ | $99.8^{+0.1}_{-0.4}$ | $100.0 \pm 0.0$ | 94.8 |
| R-2 | $99.9^{+0.1}_{-0.2}$s | $99.4^{+0.2}_{-0.3}$ | $99.8^{+0.1}_{-1.1}$ | $99.8^{+0.1}_{-0.3}$ | $94.2 \pm 0.7$ |

## C.5 Miscallaneous

Any other experiments, namely the ones on LLaMA-3 70B, LLaMa-3.1 8B, DAGER under LoRA training, or DAGER using different loss functions are included in Table 11.

## D Licenses

In our work, we use the publicly available datasets CoLA, SST-2, Rotten Tomatoes and ECHR. CoLA is licensed under the MIT license. No public licensing information was found for SST-2 and Rotten Tomatoes. Furthermore, we use ECHR under the Creative Commons Attribution-NonCommercial-ShareAlike 4.0 International (CC BY 4.0) license. For privacy concerns, ECHR has issued a statement of protection of personal data that ensures private data was handled appropriately[1].

In terms of Large Language Model architectures, we use GPT-2 under the MIT license and BERT under the Apache License. All aforementioned licenses permit our use of the underlying assets for the purposes of this paper. We obtained access to LLaMa-2 through the Llama 2 Community License Agreement[2] which permits the model's use in commercial and research settings.

Finally, we obtain the code for the LAMP and TAG attacks through the public repository for LAMP which is licensed under the Apache License 2.0.

---

[1]The statement can be found under https://www.echr.coe.int/privacy.
[2]The full license can be found under https://ai.meta.com/llama/license/.

