# OpenReview forum: "DAGER: Exact Gradient Inversion for Large Language Models"
_NeurIPS.cc/2024/Conference — NeurIPS 2024 poster_

### Official Review · Reviewer_ju5t · 2024-07-03

**Soundness:** 3
**Presentation:** 2
**Contribution:** 3
**Rating:** 6
**Confidence:** 4

**Summary:**

The authors proposed a method to recover user training data from gradients in federated learning. The key observation is if a linear network component exists in the overall neural network, the gradient of the linear component parameter can be shown as a linear combination of the input to this linear component. Taking the the first layer of the transformer, the input token embeddings can then be recovered, assuming the number of tokens involved is smaller than the dimension of the embedding space. Extending the idea to the second layer helps to recover the whole training sequence. The authors show empirically the proposed method can significantly improve the state of the art.

**Strengths:**

The idea is rather straightforward and effective. The numerical improvement over the state of the art is impressive.

**Weaknesses:**

1. The theory development is weak. The idea is rather straightforward, however, the authors need to lay down more carefully the technical assumptions. For example, is there any requirement on the loss function, i.e., MSE or cross-entropy? Are there any requirements on the differentiability? For ReLU, the loss function is sub-differentiable, and how does this impact the method?
2. The requirement that b<d needs better justification. The embedding space is usually not very large (e.g, 512, 768, or 1024). However, the number of tokens used by a single client in a single training step (or before synchronizing with the central server) can be considerably larger than this. The authors argue the condition is satisfied for reasonable input lengths and batch sizes, which I found to be less than convincing.
3. Conceptually it is not clear why the method in Algorithm 1 works so well, and some better explanation is needed. The embedded tokens are not linearly independent, and there can be significant false positives in this procedure, yet the authors observed otherwise. A more careful analysis is warranted.

**Questions:**

Some questions are as given above. Additionally,
1. On page 6, it is mentioned that the second layer filtering allows it to recover the exact 32 "starting tokens". What does it mean here "starting tokens"? Why is this an important issue to mention here?
2. Would higher layers (L>2) also help? Why is this not considered?

**Limitations:**

Yes, it was discussed in the conclusion, particularly when B is large.

---

> ### Author Rebuttal · Authors · 2024-08-07
>
> $\newcommand{\Rj}{\textcolor{green}{ju5t}}$$\newcommand{\RL}{\textcolor{red}{L4TG}}$We would like to thank reviewer $\Rj$ for the positive review and the thorough and insightful questions. We are happy they found our work effective, and the empirical results impressive, highlighting the significant improvement over state-of-the-art gradient leakage attacks. We have compiled a list of answers to any remaining concerns below.
>
> **Q1. What are the technical assumptions for DAGER? Is there any requirement on the loss function, i.e. MSE or cross-entropy? Are there any requirements on the differentiability?**
>
> We list the technical assumptions regarding the assumptions in Q2 of the general response. In short, we list the underlying assumptions and explain that they are much weaker compared to prior work. We want to emphasize that DAGER’s essential assumption is that $\frac{\partial \mathcal{L}}{\partial\bm{Q}_l}$ is full-rank. As such, the (sub-differentiable) loss function needs to non-trivially depend on all inputs, which is satisfied for any reasonable one.
>
> The full-rankness of $\frac{\partial \mathcal{L}}{\partial\bm{Q}_l}$ is also the limiting factor for any activation function. As they are applied only after the MLP at the end of a transformer block, they are less likely to directly affect $\frac{\partial \mathcal{L}}{\partial\bm{Q}_l}$. Further, Dimitrov et al. [A] have empirically demonstrated that the induced sparsity from a ReLU activation does not induce low-rankness even on the weight directly preceding it. Therefore, we reaffirm that DAGER can be applied to any reasonable activation, and that we do not depend on any particular properties.
>
> Finally, as demonstrated in **Q2** of the general response, we can indeed handle MSE loss or ReLU activations.
>
> **Q2. Why does DAGER require b<d? Is this practical given common embedding space sizes of 512 to 1024?**
> Dagger requires $b<d$ to ensure the low rankness of $\frac{\partial\mathcal{L}}{\partial\bm{W}^Q_l}$ required for our approach. However, despite this limitation, we significantly outperform every prior work by a wide margin, both in batch size and in sequence length on well-known datasets.
> Further, we want to point out that the embedding dimensionality for GPT-2 and LLaMa-2 (7B) are already 768 and 4096 respectively, (See Table 4 in the Appendix), with newer models often having even larger embedding spaces, e.g. LLaMa-3.1 (405B) has $d = 16,384$.
> Especially for these larger embedding dimensions we believe it to be quite rare to encounter this limitation in current federated learning applications. For example, in the FedLEGAL benchmark [A] with inputs of similar length to the Rotten Tomatoes dataset a batch size of up to $B=64$ poses no issue.
>
> **Q3. How do linearly dependent token embeddings affect DAGER?**
>
> It is clear that the token embeddings in a vocabulary are linearly dependent, as the vocabulary size usually exceeds the embedding size. However, this is not a concern for DAGER, as we are only interested in the embeddings present in the batch we want to reconstruct. For DAGER, it could become an issue when the embeddings of tokens in the reconstructed batch are linearly dependent. However, this is exceedingly rare for $b < d$ in practice, leading to only few false positives in the first filtering step, which we illustrate in Figure 2. Even if they do occur, they are handled correctly by our algorithm via a search procedure that is refined via span checks in the 2nd layer. With this, we are empirically able to filter all false positives, as any linear dependencies will have disappeared after the first transformer block.
>
> **Q4. On page 6, it is mentioned that the second layer filtering allows it to recover the exact 32 "starting tokens". What does it mean here "starting tokens"? Why is this an important issue to mention here?**
>
> We are happy to clarify this point! By “starting tokens” we refer to the first tokens of each of the to-be-reconstructed sequences, not any special tokens, such as [PAD] or [BOS]. The purpose of the experiment shown in Figure 2 is to demonstrate that the thresholds $\tau_1, \tau_2$ can take a wide range of values that will produce essentially the same correct solution. We mention the starting tokens in this context to simplify the presentation. The embeddings related to the 1st position are not related to any other tokens, and hence the experiment can isolate the two filtering steps without relying on any other knowledge. This is used to set the stage for the inductive progression to reconstruct the whole sequence for decoder-only models. We will add this clarification to the paper.
>
>
> **Q5. Would Filtering at higher layers (L>2) also help?**
>
> Yes, filtering at more layers would help reduce false positives. However, we observed that filtering at 2 layers is sufficient in non-noisy settings for exact reconstruction. There, further layers do not provide meaningful benefits. Therefore, we only considered two filtering layers to keep DAGER memory- and time-efficient. However, when the gradients are noisy as discussed in the response to Reviewer $\RL$’s **Q6**, additional layers provide significant benefits as they can be leveraged to average noise across different layers.
>
> **Conclusion**
>
> We hope to have been able to address all of the reviewer’s questions, are happy to answer any follow-up they might have, and look forward to their response.
>
> **References:**
>
> [A] Zhang, Zhuo, et al. "Fedlegal: The first real-world federated learning benchmark for legal nlp." *Proceedings of the 61st Annual Meeting of the Association for Computational Linguistics (Volume 1: Long Papers). 2023.*

---

> > ### Comment · Reviewer_ju5t · 2024-08-12
> >
> > Thanks for the response. I feel the authors will be able to make the expected improvement/clarification based on these questions/responses. I'll keep the score.

---

### Official Review · Reviewer_Ujeg · 2024-07-13

**Soundness:** 4
**Presentation:** 3
**Contribution:** 4
**Rating:** 7
**Confidence:** 3

**Summary:**

This is the first paper to use low rank decomposition to attack the gradient of the self-attention layer to extract information for LLM.  It also provides a fast algorithm to recover the correct token first, then the sequence.

**Strengths:**

Solid experiments and math proof. It’s a good innovative finding, especially using low rank decomposition on self-attention part.  Especially, it focuses on long sequences like 4096 and large batches for 128 on decode-base structure. The authors also provide complete ablation studies and detailed experiments in many aspects, like model size and model parameters.  Its much better than a lot of papers in the language field.

**Weaknesses:**

Notation is heavy and hard to follow. Could authors add a notation table, or input before algorithm so it is easier to follow?

Some people might argue it not use latest model but working on llama3 could be resource consuming. It might be a weaknesses of the author's work, but it could also be a future direction.

If possible, could the author discuss a little bit about the usage of quantized models and the influence of lora for fine-tuning regarding reconstruction results?

Also, could authors add some reconstructed text for comparison?

**Questions:**

see weakness above

**Limitations:**

see weakness above

---

> ### Author Rebuttal · Authors · 2024-08-07
>
> $\newcommand{\RU}{\textcolor{blue}{Ujeg}}$We would like to thank reviewer $\RU$ for their very positive review, the provided insights and helpful recommendations. We are happy they found our experiments and proofs to be solid and our method innovative. Further, we are glad that the reviewer credits the ablation studies to be complete and is particularly happy that DAGER is capable of handling long sequences and large batch sizes. Next, we address all points raised by the reviewer.
>
> **Q1. Notation is heavy and hard to follow. Could authors add a notation table, or input before the algorithm so it is easier to follow?**
>
> Yes, we provide a notation table in the rebuttal PDF and will include it in the paper.
>
> **Q2. Can you show that DAGER works on more recent models, such as LLaMa-3?**
>
> Yes, we added experiments for both LLaMa-3.1 (8B) and LLaMa-3 (70B), demonstrating that DAGER achieves perfect reconstruction for both (see Table 5 in the attached PDF). A more elaborate description of our experiments and findings can be seen in the answer to **Q1** in the main response.
>
> **Q3. Could the author discuss the usage of quantized models and the influence of LoRA for fine-tuning on reconstruction results?**
>
> Yes, we show that DAGER is directly applicable to 16-bit quantization, with 8-bit support for the setting we describe being part of active research. Further, we show that we work in the LoRA setting where the rank of the LoRA matrices is lower-bounded by $r>b$. A thorough description of our arguments and experiments can be found in **Q3** in the main response.
>
> **Q4. Could the authors add some reconstructed text for comparison?**
>
> Yes, we show examples of reconstructed text for the Rotten Tomatoes dataset on a batch size of $B=1$ in the rebuttal. We choose $B=1$ as this is a practical limitation for the baseline. As can be observed in Table 3 in the rebuttal PDF, we perfectly reconstruct the entire sentence, while the best-performing baseline manages to only get a fraction of the sentence correct.
>
> **Conclusion**
>
> We hope to have been able to address all of the reviewer’s questions, are happy to answer any follow-up they might have, and look forward to their response.

---

### Official Review · Reviewer_L4TG · 2024-07-13

**Soundness:** 4
**Presentation:** 3
**Contribution:** 3
**Rating:** 7
**Confidence:** 2

**Summary:**

In this paper, the authors propose the DAGER algorithm which leverages the low-rankness of self-attention layer gradients in order to filter out incorrect embedding vectors and recursively reconstruct true input text. DAGER works within the Centralized Federated Learning setting, where the server is honest in aggregating updates and performing model updates, but is curious to try to recover personal information from each client.

**Strengths:**

1. A very smart/practical and, to the best of my very limited knowledge in this area, novel approach to reconstruct text by disregarding all embeddings which do not align with the low rank gradients received by the server.
2. The empirical results demonstrate that this method is highly effective, and greatly outperforms all other baselines.
3. Paper is well-written, has good diagrams and figures, and is easy to follow for someone not well-versed in this area (attacks & LLMs).

**Weaknesses:**

1. I'm curious about the computational complexity of DAGER. It seems that the search space can be extremely large (even when heuristics are used), and recovering individual tokens could take awhile.
2. (minor) In the related works it would be nice to go deeper into research within the gradient leakage in the text domain. A specific breakdown of how this paper differs from the current methods.

**Questions:**

Would this be able to work within the decentralized FL environment (i.e. could a single client, acting as the aggregator, decipher other agent text)?

In the Centralized FL environment, would asynchronous training affect the performance of DAGER or render it infeasible (asynchronous, stale updates)?

There's a small error in Figure 1 in Sequence recovery (middle part, 2nd sequence), AA_ should not be in the span and BA_ should be? In the diagram the opposite is true.

How does DAGER perform against noise (DP) or malicious users? Can it be ammended to combat this?

What happens if the vocabulary size is extremely large? Won't this process take a very long time (i.e. does it scale with |V|)?
How expensive is this enumeration process? Especially if the sequence is long?

Very cool insight about size of model and how that allows gradients to be reconstructed easier because they're low rank since d is increasing so rapidly. Bigger is not always better when it comes to privacy!

**Limitations:**

Yes.

---

> ### Author Rebuttal · Authors · 2024-08-07
>
> $\newcommand{\RL}{\textcolor{red}{L4TG}}$We would like to thank reviewer $\RL$ for the very positive review and feedback. We are happy to read that the reviewer finds our work very smart, practical and novel. Further, we are glad they assessed our paper as easy to follow and our approach as highly effective. We thank the reviewer for giving us pointers on how to broaden the scale of DAGER. We now address all points raised by the reviewer.
>
> **Q1. What is the computational complexity of DAGER and does it scale to very large vocabulary sizes and long sequences?**
>
> We report both the runtimes (Table 5) and complexity (Theorem B.4) of DAGER in the Appendix of our paper. We highlight that the computational bottleneck for all practical parameters stems from the term $P^3 B^3 d^2$ or $P^4 B^3 d^2$ (in the RoPE case). As the vocabulary size is a linear component, it does not dominate the overall complexity of DAGER. We demonstrate the effectiveness on larger vocabularies by evaluating state-of-the-art models, such as LLaMa-3 70B and LLaMa-3.1 8B, which feature a vocabulary of 128,256 tokens, which is among the largest ones in existence. There, we observе only marginally increased runtimes.
>
> Please refer to **Q1** in the main response and Table 5 in the rebuttal PDF for a description of the results and requirements.
>
> **Q2. How does DAGER relate to current state-of-the-art methods in gradient leakage in the text domain?**
>
> DAGER assumes the honest-but-curious setting, where the server is non-malicious, unlike the Decepticons [28] and Panning for Gold [29] attack, meaning we do not need to change the model weights sent to clients. Most prior honest-but-curious methods, such as DLG [6], TAG [10], LAMP[11] and Jianwei et al. [31] rely on relaxing the discrete optimization problem to a continuous one that has no guarantees of convergence. One exception is APRIL [23], which analytically recovers the inputs but doesn’t scale to B>1. DAGER, however, can handle B=128. Another one is FILM [22], which requires access to the gradients of a batch at any iteration and, similarly to LAMP, requires a strong language prior, unlike DAGER.
>
> Further, we do not require knowledge of the labels for different sentences in the batch in the sentiment analysis case, in contrast to TAG and LAMP, and we work with almost any reasonable loss function (see **Q2** in the main response) unlike Flat-Chat [32].
> Finally, in contrast to APRIL, FILM and Decepticons, we do not require the gradient of the embedding layer, making our setting significantly harder. We will clarify these points in the revised version of the paper.
>
> **Q3. Would DAGER work in a decentralized FL setting with a client acting as the attacker?**
>
> While it is difficult to generalise under the vastly different decentralized FL (DFL) protocols, we attempt to provide a unified answer. The crucial part under most DFL protocols, such as IPLS [A], ProxyFL [B], or TrustedDFL [C], is that not gradients but model weights are shared between clients. We see 2 difficulties here - having multiple training steps before a model is shared, and having model weights that are aggregated across clients whose updates the attacker may not receive.
>
> The former is tightly associated with the FedAvg setting, which we have successfully applied to DAGER (see Table 4 in the rebuttal PDF). The latter is, however, a protocol-specific task, and is an interesting direction for future work.
>
> **Q4. In the Centralized FL environment, would asynchronous training affect the performance of DAGER or render it infeasible?**
>
> In centralised asynchronous FL the server and clients exchange updates asynchronously with clients sometimes creating updates based on more than one model shared by the server. Crucially, an honest-but-curious server can keep track of all updates sent to a client. This means that while a client’s gradients can be computed across several models, the models are all known to the server (as in the protocol described in [D]). Further, assuming linear aggregation, the rank of the gradient is still upper-bounded by the number of tokens used to compute it. The same formula from Theorem 3.1 still applies with $\mathbf{X}$’s coming from different models. To this end, the server will need to apply DAGER for tokens propagated through all possible client models.
>
> **Q5. There's a small error in Figure 1 - AA_ should not be in the span and BA_ should be.**
>
> Yes, thank you for spotting it! We will fix it.
>
> **Q6. How does DAGER perform against noise (DP) or malicious users? Can it be amended to combat this?**
>
> It has been shown [E] that DP provides provable guarantees for protecting privacy. Nevertheless, we provide promising initial results, where we use deeper layers to mitigate the effect of noise. The crucial assumption here is that noise is independent across layers. We apply DAGER on different noise levels with $B=1$ on the Rotten Tomatoes dataset utilising the GPT-2 model. The results can be found in Table 2 in the rebuttal PDF. We believe that there are numerous improvements one could make, but leave these for future work.
>
> It is unclear to us what the reviewer refers to as “malicious user” and how input reconstruction could help there. We kindly ask for some clarification and are more than happy to address the question.
>
> **Conclusion**
>
> We hope to have been able to address all of the reviewer’s questions, are happy to answer any follow-up they might have, and look forward to their response.
>
> **References:**
>
> [A] Pappas, C., et al. "Ipls: A framework for decentralized federated learning." *2021 IFIP.*
>
> [B] Kalra, S., et al. "Proxyfl: decentralized federated learning through proxy model sharing." *(2021).*
>
> [C] Gholami, A., et al. "Trusted decentralized federated learning." *CCNC, 2022.*
>
> [D] Chen, Y., et al. "Asynchronous online federated learning for edge devices with non-iid data." *Big Data. 2020.*
>
> [E] Abadi, M., et al. "Deep learning with differential privacy." *ACM SIGSAC. 2016.*

---

### Author Rebuttal · Authors · 2024-08-07

$\newcommand{\RL}{\textcolor{red}{L4TG}}$$\newcommand{\RU}{\textcolor{blue}{Ujeg}}$$\newcommand{\Rj}{\textcolor{green}{ju5t}}$$\newcommand{\bm}[1]{\mathbf{#1}}$$\newcommand{\dl}{{\partial\mathcal{L}}}$$\newcommand{\dldz}{\frac{\dl}{\partial\bm{Q}_l}}$We would like to thank the reviewers for their very positive feedback and valuable input which will help us improve our paper. We are pleased the reviewers found our improvement over prior work very significant, our method effective ($\RL$, $\RU$, $\Rj$), and our findings innovative ($\RL$, $\RU$).

Based on the reviewers' suggestions, we conducted a range of additional experiments and reported results in the rebuttal PDF. Below, we will address the most important and shared points the reviewers raised, before addressing all remaining individual questions in reviewer-specific responses.

**Q1: Does DAGER scale to newer and larger models and larger vocabularies? ($\RL$, $\RU$)**

Yes. We applied DAGER on the much larger LLaMa-3 70B and very recent LLaMa-3.1 8B, showing results in Table 5 of the rebuttal PDF, and observed that it achieves outstanding R1 and R2 scores of over 99%.

In more detail, we applied DAGER on 100 batches from Rotten Tomatoes using the 16-bit quantized models and batch sizes of $B=1$ for LLaMa-3 70B and $B=32$ for LLaMa-3.1 8B. Comparing total runtimes to LLaMa-2 7B (39.5h on 1 A100), we observe that LLaMa-3.1 8B (41.4h on 1 A100) takes only marginally longer despite having almost double the vocabulary size (128,256 tokens), highlighting that vocabulary size has little effect on the runtime of DAGER. For the 10 times larger LLaMa-3 70B, a four times longer runtime of 167.4h and 8 NVIDIA A100 GPUs were needed due to the memory requirements of computing gradients for such large models.

These results confirm the generality and scalability of our algorithm to large models and vocabulary sizes and its applicability to state-of-the-art LLMs.

**Q2: What assumptions does DAGER make? ($\RL$, $\Rj$)**

DAGER makes three assumptions:
 - We assume that $\dldz$ is full-rank.
 - We require the total number of tokens $b$ in the batch to be smaller than the embedding dimension $d$, ensuring that $\frac{\dl}{\partial\bm{W}^Q_l}$ is of low-rank
 - We assume a known discrete set of possible inputs to the model, i.e. its vocabulary.

Importantly, DAGER does not assume any prior knowledge of the labels or the lengths of each sequence in the batch nor access to the gradients of the embedding layers which have been shown to leak significant information [23]. Further, we require no language priors and operate under the honest-but-curious setting which does not allow malicious changes to model weights. Finally, sub-differentiability is sufficient for applying DAGER.

In practice, DAGER requires much fewer assumptions than prior works while being successful in a variety of common LLM tasks, e.g., next-token prediction and sentence classification. While these tasks use the cross-entropy loss, DAGER can be applied to any loss function that non-trivially depends on every input token and thus ensures the full-rankness of $\dldz$.

To confirm its generality, we apply DAGER with a Frobenius norm-based loss and ReLU activation functions. We use a custom loss function $\mathcal{L}(s_1, s_2, …, s_P) = ||{\bm{f}^L(s_1, s_2, …, s_P)}||_F$, where $||.||_F$ is the Frobenius norm, which is equivalent to an MSE loss with $\bm{0}$ as a target vector. We report the results of applying DAGER with these modifications on Rotten Tomatoes using GPT-2 with $B=16$ in Table 5 in the rebuttal PDF, achieving R1 and R2 > 99% in both cases.

**Q3. Can DAGER be applied to quantized models or during LoRA fine-tuning?** ($\RU$)

Yes, DAGER can be applied both to quantized models and during LoRA fine-tuning.

For both Llama3 70B and Llama 3.1 8B at 16-bit quantization, we observed excellent R1 and R2 >99% after adapting the threshold values $\tau_1$ and $\tau_2$ to the increased numerical instabilities (see Table 5 in the attached PDF). While we expect DAGER to be applicable to 8-bit quantization, Hugging Face’s `transformers` package, which we use in our implementation, currently lacks support for full-model FP8 training, preventing us from confirming this empirically during the rebuttal period. We note that prior work [A] has shown that training with (partial) 8-bit quantization requires some operations to be done in at least 16-bit precision, making a quick re-implementation out of scope for this rebuttal.

We further confirm that DAGER can be directly applied to LoRA training as follows. Using the low-rank representation of the weight matrix $\bm{W}=\bm{W}_0 + \bm{AB}$, with gradient updates performed on $\bm{A}\in\mathbb{R}^{d\times r}, \bm{B}\in\mathbb{R}^{r\times d}$, we can apply Theorem 5.1 of DAGER to $\frac{\dl}{\partial\bm{A}} = \frac{\dl}{\partial \bm{X}\bm{A}}\bm{X}^T$. Assuming that $\frac{\dl}{\partial \bm{X}\bm{A}}$ is full-rank and that b<r, our work is directly applicable. In practice, LoRA finetuning typically initializes $\bm{W} = \bm{W}_0$ and $\bm{B}$ to only contain zeroes which reduce the rank of $\frac{\dl}{\partial\bm{A}}$ for the first few optimization steps. We therefore train LLaMa-3.1 8B on the Rotten Tomatoes using a batch size of 4 with $r=256$ (following [B]) for 3 epochs before applying DAGER. We report results in Table 5 of the rebuttal PDF and observe an excellent R1 and R2 of about 95%.

**Conclusion**

We hope we were able to address the reviewers’ questions and concerns and look forward to an active discussion phase.

**References:**

[A] Xi, Haocheng, et al. (2024). "Jetfire: Efficient and Accurate Transformer Pretraining with INT8 Data Flow and Per-Block Quantization." *arXiv:2403.12422*

[B] Biderman, Dan, et al. "Lora learns less and forgets less." *arXiv:2405.09673 (2024).*

---

### Decision · Program_Chairs · 2024-09-25

**Decision:**

Accept (poster)

**Comment:**

DAGER is a method to reconstruct input text for Centralized Federated Learning. In this problem there is a federated learning aggregating server (that is honest but curious) that is aggregating gradient updates and performs the model weight updates.
The proposed method leverages the low-rankness of self-attention gradients to find the correct embeddings and reconstruct training data.
This is is a narrow but interesting problem. Overall the reviewers were positive (but not enthusiastic) about this manuscript. The authors did a good job in the rebuttal and the paper should be accepted as a poster.